# Limits on information transduction through amplitude and frequency regulation of transcription factor activity

Anders S Hansen[1,2], Erin K O'Shea[1,2,3]*

[1]Department of Chemistry and Chemical Biology, Howard Hughes Medical Institute, Harvard University, Cambridge, United States; [2]Faculty of Arts and Sciences Center for Systems Biology, Harvard University, Cambridge, United States; [3]Department of Molecular and Cellular Biology, Harvard University, Cambridge, United States

**Abstract** Signaling pathways often transmit multiple signals through a single shared transcription factor (TF) and encode signal information by differentially regulating TF dynamics. However, signal information will be lost unless it can be reliably decoded by downstream genes. To understand the limits on dynamic information transduction, we apply information theory to quantify how much gene expression information the yeast TF Msn2 can transduce to target genes in the amplitude or frequency of its activation dynamics. We find that although the amount of information transmitted by Msn2 to single target genes is limited, information transduction can be increased by modulating promoter *cis*-elements or by integrating information from multiple genes. By correcting for extrinsic noise, we estimate an upper bound on information transduction. Overall, we find that information transduction through amplitude and frequency regulation of Msn2 is limited to error-free transduction of signal identity, but not signal intensity information.

## Introduction

Cellular signaling pathways often exhibit a bowtie topology (*Csete and Doyle, 2004*): multiple distinct signal inputs converge on a single master regulator, typically a transcription factor (TF), which then controls the expression of partially overlapping sets of downstream target genes. This raises two general questions: first, how can the cell encode information about different signals in the activity of a single master TF? Second, can this information be decoded by target genes to elicit a specific output for each input?

One way the cell can encode signal information is by regulating the activation dynamics of a single master TF (*Figure 1A*). For example, p53, a tumor suppressor TF, exhibits an intensity-dependent number of nuclear pulses in response to γ-radiation, but a sustained pulse of nuclear localization with intensity-dependent amplitude during UV-radiation (*Lahav et al., 2004*; *Batchelor et al., 2011*). Akin to p53, the yeast multi-stress response TF Msn2 exhibits short pulses of nuclear localization with intensity-dependent frequency under glucose limitation, but sustained nuclear localization with intensity-dependent amplitude under oxidative stress (*Hao et al., 2013*; *Hao and O'Shea, 2012*; *Jacquet et al., 2003*; *Petrenko et al., 2013*). Thus, p53 and Msn2 dynamics encode both signal identity and signal intensity. Beyond p53 and Msn2, amplitude- or frequency encoding of signal identity and intensity information is conserved throughout eukaryotic signaling pathways (see also *Berridge et al., 2000*; *Werner et al., 2005*; *Cai et al., 2008*; *Warmflash et al., 2012*; *Albeck et al., 2013*; *Aoki et al., 2013*; *Imayoshi et al., 2013*; *Dalal et al., 2014*; *Harima et al., 2014*). Such encoding of signal identity and intensity information in TF activation dynamics has led to the hypothesis that TF target genes can reliably decode this dynamical information to elicit distinct gene expression programs with

*For correspondence:
Erin_Oshea@harvard.edu

**Competing interests:**
See page 16

**eLife digest** The way that a cell responds to an external stimulus is governed by a sequence of events called a signalling pathway. While cells are exposed to a wide range of external stimuli—such as different types of chemicals and different forms of radiation—the last stage of the signalling pathway usually involves a gene being expressed as a protein or some other gene product. The amount of protein that is produced depends on the intensity of the signal that reaches the end of the pathway.

Proteins called transcription factors have an important role in this gene expression stage, and it is quite common for several signalling pathways to pass through the same transcription factor. How does the cell ensure that the information travelling along a particular pathway reaches the relevant gene and that the correct level of gene expression takes place?

Biologists have been using information theory—a set of ideas developed by computer scientists and engineers—to understand signalling pathways at a fundamental level. It turns out that just as radio stations can broadcast on FM (which is short for frequency modulation) or AM (amplitude modulation), cells can do something similar. Msn2 is a transcription factor that is found in yeast: when the supply of glucose to the yeast cells is limited, Msn2 becomes active in short bursts, with the *frequency* of the bursts depending on the severity of the glucose shortage (which is similar to FM radio). However, when the yeast cells are exposed to chemicals that cause oxidative stress, Msn2 becomes active for prolonged periods, with the *amplitude* of this activity depending on level of oxidative stress (similar to AM radio).

In the language of information theory, the behaviour of Msn2 encodes two types of information: information about identity (short bursts of activity, or FM, mean that there is a shortage of glucose; sustained bursts, or AM, mean that the cell is experiencing oxidative stress), and information about intensity (that is, information about the severity of the glucose shortage or the level of oxidative stress). But how much of this information is transmitted to the relevant genes?

Hansen and O'Shea have used a combination of experiment and information theory to explore this question. For both the AM and FM cases, it is found that the cell can transmit the identity information but not the intensity information. However, the amount of information transmitted can be increased by having multiple copies of the same gene, by combining information from more than one gene, or by modifying a region of DNA called a promoter that is involved in the regulation of genes.

Finally, unlike radio broadcasting, where FM is generally favoured over AM, Hansen and O'Shea find that AM signalling is more reliable than FM signalling in cells. In the future, it will be a priority to investigate whether these results apply more generally beyond the Msn2 system in yeast.

fine-tuned expression levels (*Figure 1A*) (*Behar et al., 2007*; *Behar and Hoffmann, 2010*; *de Ronde and ten Wolde, 2014*; *Hansen and O'Shea, 2013*; *Levine et al., 2013*; *Purvis and Lahav, 2013*; *Yosef and Regev, 2011*).

However, non-genetic cell-to-cell variability (noise) in gene expression limits the fidelity with which information can be decoded by TF target genes (*Coulon et al., 2013*; *Sanchez and Golding, 2013*). This is important because the capacity of any signaling pathway for information transduction is limited by the capacity of its weakest node or bottleneck (*Cover and Thomas, 2006*). Thus, even though information can reliably be encoded in TF activation dynamics (*Selimkhanov et al., 2014*), this information will be lost unless genes can reliably decode it. We therefore focus on the response of single genes and ask: can cells reliably transmit both signal identity and intensity information in the amplitude and frequency of TFs to target genes in the presence of biochemical noise? In other words, what are the limits on amplitude- and frequency-mediated information transduction? We investigate this by applying tools from information theory to quantify how much of the information (in bits) encoded in the amplitude and frequency of a TF can be transmitted through gene promoters to fine-tune the gene expression level.

Originally developed by Claude Shannon for communication systems (*Shannon, 1948*), information theory has recently been applied to cell signaling (reviewed in *Tkacik and Walczak, 2011*; *Waltermann and Klipp, 2011*; *Nemenman, 2012*; *Rhee et al., 2012*; *Bowsher and Swain, 2014*;

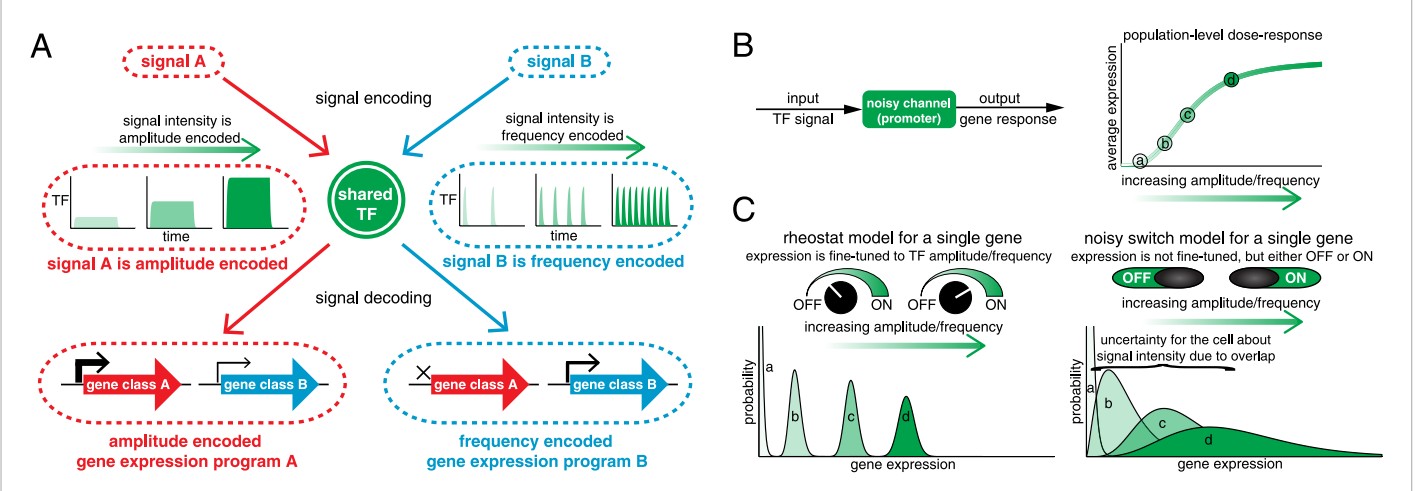

**Figure 1**. Encoding and transmitting signal identity and intensity information in the dynamics of a single transcription factor (TF). (**A**) Different signals (e. g., stress or ligand exposure) can be encoded in the dynamics of a single TF. Signal identity is encoded in the type of TF dynamics: a sustained pulse (signal A) or nuclear pulsing (signal B). Signal intensity (e.g., ligand concentration) is encoded in the amplitude for signal A, but in the frequency for signal B. Different dynamical patterns of TF activation can activate distinct, but specific, downstream gene expression programs. (**B**) Applying an information theoretic framework to cell signaling, a gene promoter can be considered a channel. A graded population-level dose–response belies the complexity of the single-cell response: it shows the mean expression at points a, b, c, and d, but not the width or variance of their distributions. (**C**) Two extreme models. In the 'rheostat model', signal intensity information encoded in the frequency or amplitude of a TF leads to non-overlapping gene expression distributions (a, b, c, and d). Thus, by reading the gene expression output the cell can accurately determine the input signal intensity and high information transmission is achieved. Conversely, in the 'noisy switch model', as a consequence of overlapping gene expression distributions (a, b, c, and d) information about signal intensity is permanently lost: the cell can distinguish ON/OFF (signal identity), but the expression of a target gene cannot be fine-tuned to the stress intensity.

*Levchenko and Nemenman, 2014*; *Mc Mahon et al., 2014*). Mutual information quantifies how much information an output can carry about an input across a noisy channel (*Figure 1B*). Mathematically, information is quantified in bits: to resolve two different signal intensities without error requires at least 1 bit of information, to resolve four different signal intensities without error requires at least 2 bits of information and so forth. However, 1 bit of information does not guarantee that two intensities can be distinguished without error. Similarly, 1 bit may allow multiple intensities to be distinguished, albeit with some associated error (*Bowsher and Swain, 2014*). As an example of how information theory can be applied, consider a dose–response relationship (*Figure 1B*). A graded population-level dose–response can belie the complexity of the single-cell response (*Ferrell and Machleder, 1998*). For example, if different TF amplitudes or frequencies lead to distinguishable gene expression outputs (points a, b, c and d), signal intensity information is accurately transmitted and the cell can fine-tune the expression of stress genes to the stress intensity like a 'rheostat' (*Figure 1C*, rheostat model). However, biochemical noise can degrade signal information: if gene expression outputs are no longer resolvable, the cell can no longer fine-tune the expression level of stress genes to stress intensity (*Figure 1C*, noisy switch model). In the noisy switch model, the cell can distinguish no stimulus (point a, OFF) from maximal stimulus (point d, ON)—but intermediate stimuli (points b and c) cannot reliably be distinguished based on the gene expression output and signal intensity information has been lost (*Figure 1C*). Information theory provides a framework for capturing and quantifying these differences. Thus, we can distinguish these two models by measuring information transduction by promoters: the noisy switch model requires ~1 bit, whereas the rheostat model requires substantially higher mutual information.

Previous applications of information theory have been theoretical (*Ziv et al., 2007*; *Tostevin and ten Wolde, 2009*; *Lestas et al., 2010*; *de Ronde et al., 2011*; *Bowsher and Swain, 2012*; *Rieckh and Tkacik, 2014*) or have focused on upstream signaling and development (*Gregor et al., 2007*; *Tostevin et al., 2007*; *Skerker et al., 2008*; *Tkacik et al., 2008, 2009*; *Mehta et al., 2009*; *Cheong et al., 2011*; *Dubuis et al., 2013*; *Uda et al., 2013*; *Selimkhanov et al., 2014*; *Voliotis et al., 2014*).

However, despite gene expression being the final bottleneck in cell signaling, gene expression has received little attention (*Uda et al., 2013*). Estimating an upper limit on the information transduction capacity of a gene has not previously been possible due to extrinsic noise: even when studying genetically identical single cells, the cells can exhibit non-genetic differences, that is, in cell cycle phase or variability in TF concentration, which means the measured mutual information will be an underestimate (*Elowitz et al., 2002*; *Toettcher et al., 2013*). Here, we overcome this limitation through a combined experimental and theoretical approach that corrects for extrinsic noise and allows us to estimate an upper limit on the information transduction capacity of individual Msn2 target genes.

We combine high-throughput microfluidics to control the amplitude and frequency of Msn2 nuclear translocation with information theory to determine the information transduction capacity of Msn2 target genes. We find that Msn2 target genes can transduce just over 1 bit of information, consistent with the 'noisy switch model'. Although individual Msn2 target genes can only transduce little information, we illustrate how the cell can improve information transduction capacity by modulating promoter *cis*-elements, by integrating the response of more than one gene, or by having multiple copies of the same gene. We show that more information can be transduced through amplitude than through frequency modulation (FM) of Msn2 activation dynamics. Nevertheless, while previous studies have shown that significant amounts of information can be encoded in TF activation dynamics (*Selimkhanov et al., 2014*), we find that noise in the decoding step severely limits information transduction. Specifically, our results indicate that information about signal identity, but not signal intensity, can be transmitted nearly without error in the amplitude and frequency of Msn2 and decoded by Msn2-responsive promoters.

## Results

### Quantifying information transduction using information theory

Information theory quantifies information transduction across a channel between a signal and a response (*Shannon, 1948*; *Cover and Thomas, 2006*). If a channel is noisy, a given signal input will give rise to a distribution of response outputs. This represents a loss of information since the signal input can no longer reliably be learned from observing the response output (*Figure 1B–C*). A 'black-box'-framework, information theory was originally developed for telecommunication channels, but it can also be applied to other 'channels' such as gene promoters or cell signaling pathways provided that the signal input (here amplitude or frequency of Msn2 activation) can be precisely controlled and the response output distribution precisely measured (here single-cell gene expression). Mutual information, $MI(R;S)$, measured in bits, quantifies the amount of information about the signal input ($S$) that can be obtained by observing the response output ($R$) and, given discretized data, is defined as:

$$MI(R;S) = \sum_{i,j} p(R_i, S_j) \log_2 \left( \frac{p(R_i, S_j)}{p(R_i)p(S_j)} \right). \tag{1}$$

The response distribution, $p(R)$, is the experimentally measured distribution of gene expression output. The signal distribution, $p(S)$, is the relative probability of each Msn2 amplitude or frequency. Since $MI(R;S)$ depends on $p(S)$ and since $p(S)$, that is, how often a cell might be exposed to a particular intensity of oxidative stress, is unknowable, hereafter we consider the maximal mutual information, $I$ ($I(R;S) = \max_{p(S)}[MI(R;S)]$) which is the maximal amount of information that can be transduced through a 'promoter channel'. $I$ can be thought of as a channel capacity, though a gene promoter is effectively a 'single-use' channel and $I$ therefore has units of bits, whereas messages are sent repeatedly through a Shannon channel and, accordingly, the channel capacity has units of bits/s ([*Bowsher and Swain, 2014*]; a detailed discussion is given in *Supplementary file 2*).

### Natural Msn2 target genes have low information transduction capacities

To measure how much information Msn2 target genes can transduce, we took advantage of a pharmacological method for controlling Msn2 nuclear localization using a small molecule, 1-NM-PP1, (*Bishop et al., 2000*; *Hao and O'Shea, 2012*; *Zaman et al., 2009*) and high-throughput microfluidics coupled to quantitative time-lapse microscopy (*Hansen et al., 2015*; *Hansen and O'Shea, 2013*). With this setup (*Figure 2—figure supplement 1A*; *Video 1*), we can control and measure the amplitude and frequency of activation of an Msn2-mCherry fusion protein over time and generate single-cell traces

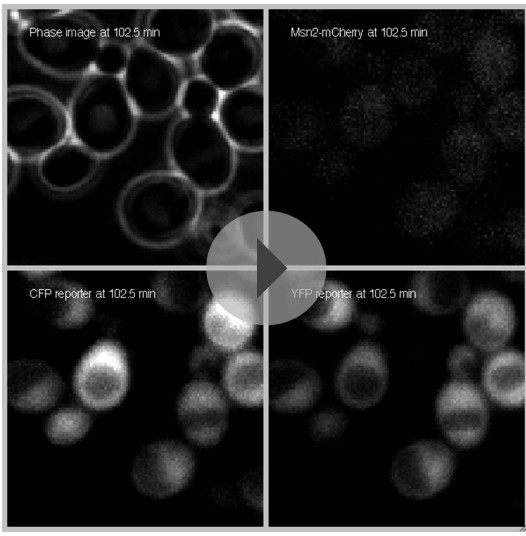

**Video 1.** A typical experiment. Mut B cells were grown in a microfluidic device and exposed to six 5-min Msn2 pulses separated by 10 min and phase contrast (top left), Msn2-mCherry (top right), CFP (bottom left), and YFP (bottom right) reporter expression monitored. Video 1 consists of 64 frames at 2.5 min resolution and images have been compressed, cropped, and contrast adjusted, but not corrected for photobleaching.

that mimic the natural Msn2 dynamics under oxidative stress (a sustained nuclear pulse with signal intensity–dependent amplitude; *Figure 2A*) and glucose limitation (short pulses with signal intensity–dependent frequency; *Figure 2B*) (*Hao and O'Shea, 2012*; *Petrenko et al., 2013*). To measure stress-relevant gene expression, we use dual cyan and yellow fluorescent protein (CFP/YFP) reporters and focus on two specific Msn2 target genes: *HXK1*, which is induced under glucose limitation (*Herrero et al., 1995*) and *SIP18*, which is induced in response to oxidative stress (*Rodriguez-Porrata et al., 2012*). Using this setup, we have previously shown that, at the population level, individual genes differentially decode Msn2 dynamics (*Hansen and O'Shea, 2013*; *Hao and O'Shea, 2012*): oscillatory Msn2 activation induces gene class B (e. g., *HXK1*) without inducing gene class A (e.g., *SIP18*), whereas sustained Msn2 activation preferentially induces gene class A (*Figure 1A*). Thus, this represents an ideal setup for studying promoter decoding of Msn2 dynamics in single cells, which enables us to quantify information transduction.

To measure information transduction through the *HXK1* and *SIP18* promoters with respect to amplitude modulation ($I_{AM}$), we exposed thousands of cells to increasing amplitudes of a 70 min Msn2 pulse to mimic oxidative stress, measured the single-cell distribution of responses for each amplitude with minimal measurement noise (*Figure 2—figure supplement 2*), and determined the population-averaged dose–response (*Figure 2A*; all raw single-cell data are available online as *Supplementary file 1* and in (*Hansen and O'Shea, 2015*); see also *Figure 2—figure supplement 1B*). We quantify gene expression as the maximal YFP concentration after the YFP time-trace has reached a plateau ('Materials and Methods'). Surprisingly, for both *HXK1* and *SIP18*, $I_{AM}$ was 1.2–1.3 bits—enough to distinguish ON from OFF without error (the 'no Msn2 input' and the 'brown' distributions are clearly distinguishable; *Figure 2A*), but with limited ability to distinguish signal intensities. One way to think about this result is to ask, given the *HXK1* YFP expression output, how much information does that provide about the input amplitude? For example, considering the *HXK1* AM histograms in *Figure 2A*, for most YFP outputs the cell can exclude the 'no Msn2 input' condition, but appears to be unable to discern which of the other amplitudes it was exposed to without a high error rate. Consequently, *HXK1* and *SIP18* can distinguish no stress from high oxidative stress (high Msn2 amplitude) without error, but cannot accurately transmit information about stress intensity.

Next, we measured information transduction of *HXK1* and *SIP18* with respect to frequency modulation ($I_{FM}$) using 5-min Msn2 pulses at frequencies similar to those observed under glucose limitation (*Figure 2B*). Even though *HXK1* is physiologically induced during Msn2 pulsing, $I_{FM}$ was only 1.11 bits—again enough for distinguishing ON from OFF essentially without error like a 'noisy switch', but insufficient to accurately fine-tune the *HXK1* expression level to each Msn2 frequency like a 'rheostat'. *SIP18*, required only under oxidative stress, largely filters out Msn2 pulsing and therefore has a negligible $I_{FM}$.

## The promoter information transduction capacity is tunable and can be increased for natural Msn2 target genes

It is generally assumed that gene expression levels are fine-tuned (*de Nadal et al., 2011*), but the very low $I_{AM}$ and $I_{FM}$ of *HXK1* and *SIP18* are incompatible with this idea. One possibility is that mutual information for promoters is biophysically constrained to ~1.0–1.3 bit, but another possibility is that

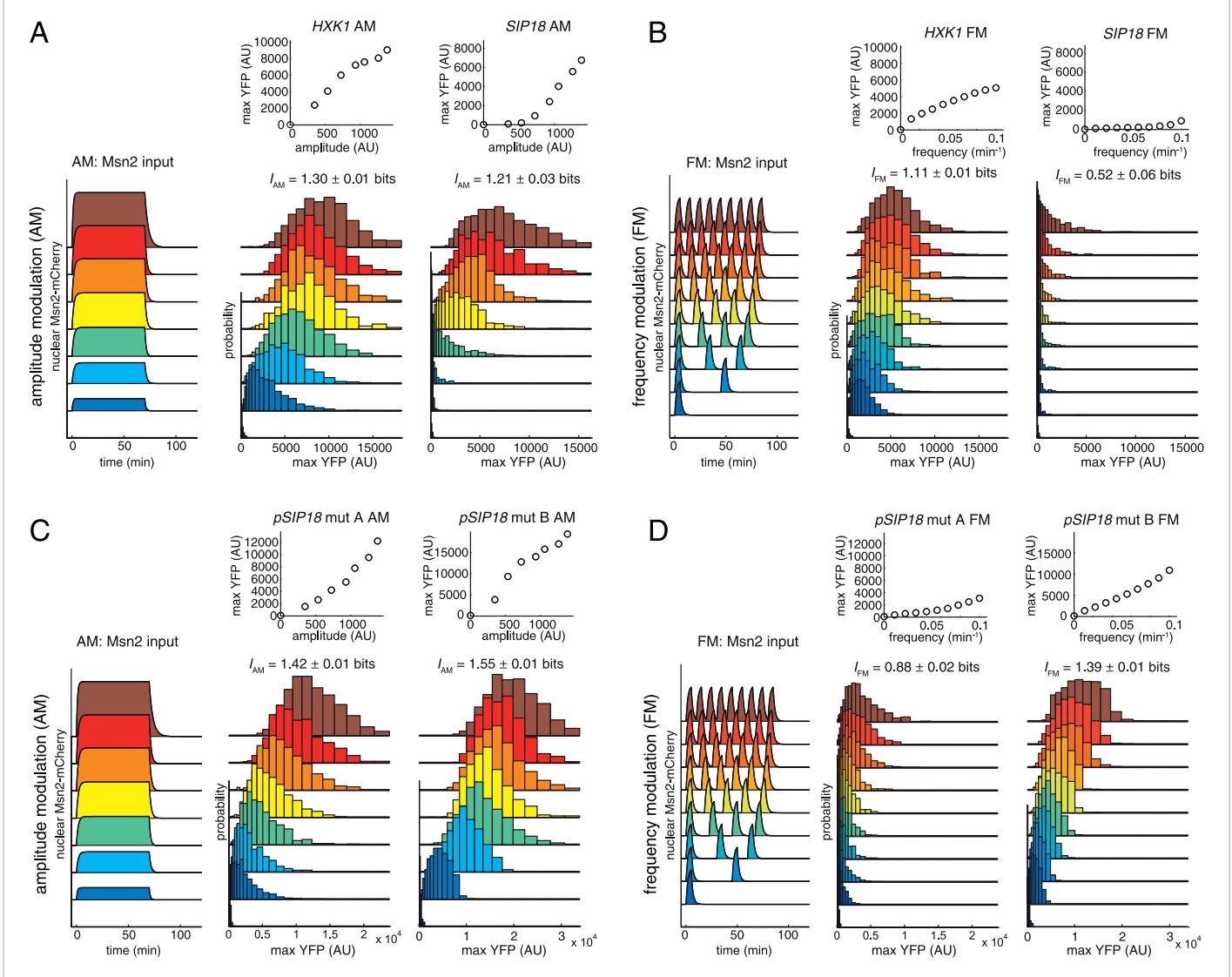

**Figure 2.** Information transduction by promoters with respect to amplitude and frequency modulation. (**A**) Cells containing either the *hxk1*::YFP or *sip18*:: YFP reporter were exposed to either no activation or a 70-min pulse of seven increasing amplitudes from ca. 25% (100 nM 1-NM-PP1) to 100% (3 μM 1-NM-PP1) of maximal Msn2-mCherry nuclear localization and single-cell gene expression monitored. For each single-cell time-trace, YFP concentration is converted to a scalar by taking the maximal YFP value after smoothing. For each Msn2-mCherry input (a fit to the raw data is shown on the left (AM: Msn2 input)), the gene expression distribution is plotted as a histogram of the same color on the right for *HXK1* and *SIP18*. The population-averaged dose–response (top) is obtained by calculating the YFP histogram mean for each Msn2 input condition. (**B**) Cells containing either the *hxk1*::YFP or *sip18*:: YFP reporter were exposed to either no activation or from one to nine 5-min pulses of Msn2-mCherry nuclear localization (ca. 75% of maximal nuclear Msn2-mCherry, 690 nM 1-NM-PP1) at increasing frequency. All calculations were performed as in (**A**). (**C**) Cells containing either the *pSIP18* mut A::YFP reporter or the *pSIP18* mut B::YFP reporter were exposed to amplitude modulation (AM) as in (**A**). (**D**) Cells containing either the *pSIP18* mut A::YFP reporter or the *pSIP18* mut B::YFP reporter were exposed to frequency modulation (FM) as in (**B**). Maximal mutual information, *I*, and its error are calculated as described in *Supplementary file 2*. Full details on data processing are given in 'Materials and Methods'. Each plot of an Msn2 input pulse and YFP expression is based on data from ca. 1000 cells from at least three replicates. All raw single-cell time-lapse microscopy source data for *HXK1* (15,259 cells), *SIP18* (21,242 cells), *pSIP18* mut A (18,203 cells), and *pSIP18* mut B (17,655 cells) for this Figure are available online as *Supplementary file 1* and in (*Hansen and O'Shea, 2015*).

The following figure supplements are available for figure 2:

**Figure supplement 1**. How time-lapse data are converted to histograms and promoter maps and noise data.

**Figure supplement 2**. Data processing and control of measurement noise.

*HXK1* and *SIP18* are not optimized for AM- and FM-mediated information transduction. To investigate this and explore the relationship between promoter *cis*-elements and information transduction, we focused on *SIP18*, which has the lowest *I* and suffers from high gene expression noise (*Figure 2—figure supplement 1D*), and asked if altering promoter architecture could improve information transduction. We removed the two functional Msn2 binding sites in the *SIP18* promoter and added three and four new binding sites in the nucleosome-free region closer to the transcription start site (promoter architecture maps are shown in *Figure 2—figure supplement 1C*) to generate *pSIP18* mut A and *pSIP18* mut B, which differ from the wild-type *SIP18* promoter by 14 and 18 nucleotides, respectively. We then repeated the experiments for mut A and mut B to measure their $I_{AM}$ and $I_{FM}$.

With respect to AM, both mutants had significantly higher $I_{AM}$ of 1.42 bits (mut A) and 1.55 bits (mut B) (*Figure 2C*). We attribute this increase to a combination of three factors: a more linear dose–response, a higher dynamic range, and significantly lower gene expression noise (*Figure 2—figure supplement 1D*).

The wild-type *SIP18* promoter filters out oscillatory input and therefore has a negligible $I_{FM}$. In contrast, with respect to FM mut A shows a slightly higher $I_{FM}$ of 0.88 bits and mut B a significantly higher $I_{FM}$ of 1.39 bits (*Figure 2D*). Notably, although *HXK1* presumably evolved to decode Msn2 pulsing, as is observed under glucose limitation, mut B now shows a higher $I_{FM}$ than even *HXK1*. Although *I* could be different for natural Msn2 dynamics (*Hao and O'Shea, 2012*), these results show that for the AM and FM signals studied here, natural Msn2 target genes are not optimized for information transduction and do not have their maximal *I* even though promoters with higher $I_{AM/FM}$ are only a few mutations away. Furthermore, $I_{AM}$ exceeds $I_{FM}$ for all four promoters, which shows that, at least in these four cases, transmitting gene expression information in the amplitude of TF activation dynamics is more reliable than transmitting it in the frequency. Thus, the promoter information transduction capacity is tunable in *cis*: by modulating Msn2 binding sites, we can control both how a promoter decodes Msn2 dynamics and how much information it can transmit.

## Estimating the intrinsic information transduction capacity of promoters

Natural Msn2 target promoters appear to have $I \leq 1.3$ bits. Thus, we observe high information loss during gene expression. Information loss comes from two sources: gene-intrinsic and gene-extrinsic noise (*Elowitz et al., 2002*). Intrinsic noise originates from the inherently stochastic nature of biochemical reactions, such as stochastic binding of Msn2 at individual promoters. Information loss due to intrinsic noise is therefore unavoidable for the cell. Extrinsic noise comes from the intracellular environment, which may differ between cells in a population. Even though we consider genetically identical cells grown in a microfluidic chemostat, the cell population could exhibit non-genetic differences in cell-cycle phase and Msn2 abundance or dynamics, etc. This could cause the dose–response to be different between single cells (*Figure 3A*), as was observed in a recent study on Ras/ERK signaling (*Toettcher et al., 2013*). For example, a cell with a higher-than-average Msn2 abundance might show higher gene expression. When we carefully quantify Msn2-mCherry dynamics, we observe loss of information between the microfluidic 1-NM-PP1 input and nuclear Msn2 due to variability in Msn2 abundance between cells (*Figure 3—figure supplement 1*). Likewise, the cell cycle is a major source of extrinsic gene expression noise (*Zopf et al., 2013*). Therefore, measuring mutual information in a cell population subject to extrinsic noise, as we did in *Figure 2*, underestimates the intrinsic information transduction capacity of a promoter.

Although it is in principle possible to correct for cell cycle phase, Msn2 abundance and other gene-extrinsic factors individually, it is impossible to correct for all factors. To overcome this limitation and estimate the intrinsic *I* ($I_{int}$), we developed a method based on the dual-reporter approach (*Elowitz et al., 2002*; *Swain et al., 2002*; *Hilfinger and Paulsson, 2011*). By having two gene expression reporters in diploid cells on homologous chromosomes that differ only by their color (CFP and YFP) but share the same intracellular environment, the extent to which they co-vary in the same cell allows us to infer how much gene-extrinsic factors, such as cell-cycle phase and Msn2 variability, and so on. contribute altogether (extrinsic noise), without having to specify each factor. Or phrased differently, if the dose–response is shifted in a cell, both the CFP and YFP reporter will be affected in a correlated manner and their covariance allows us to quantify this (*Figure 3A*). Therefore, we developed an algorithm that uses the CFP/YFP covariance to estimate what the intrinsic *I* ($I_{int}$) would have been in the absence of extrinsic noise. Briefly, our algorithm takes the following steps (*Figure 3B*): First, the raw YFP histogram is fitted to a gamma distribution (YFP $\sim \Gamma(a, b)$). Second, the extrinsic component (covariance) of the total variance is determined ($\sigma_{ext}^2 = \langle CFP \cdot YFP \rangle - \langle CFP \rangle \langle YFP \rangle$).

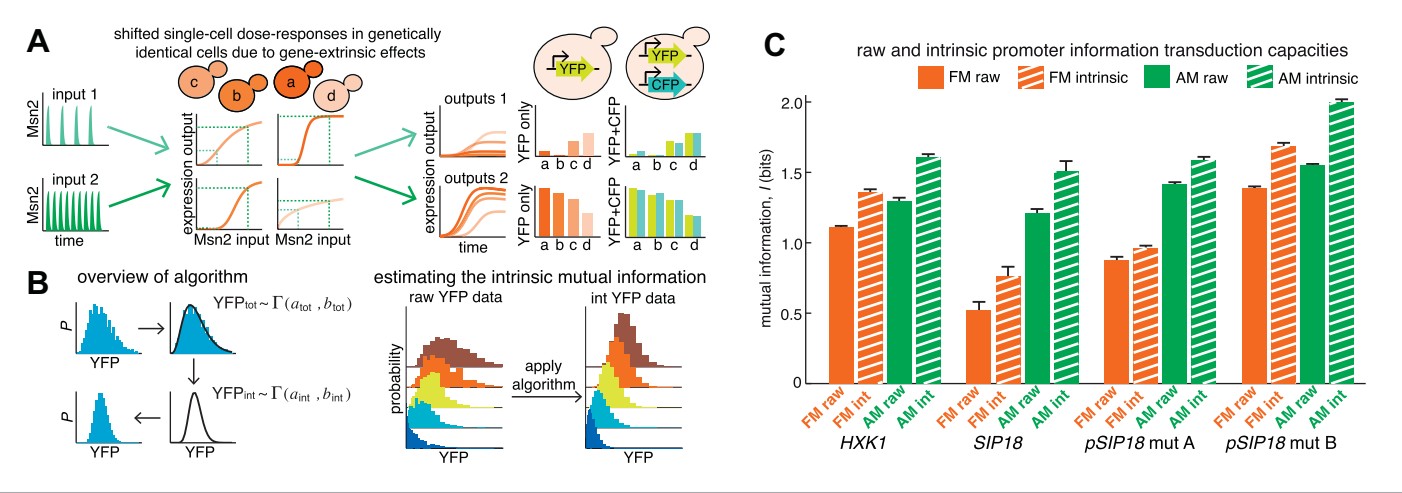

**Figure 3**. An algorithm for estimating intrinsic mutual information. (**A**) Genetically identical cells can have shifted single-cell dose-responses due to gene-extrinsic effects, such as variation in Msn2 abundance and cell cycle phase. Measuring the response of a single reporter (YFP) therefore underestimates mutual information. By introducing an additional reporter (CFP), we can distinguish extrinsic noise such as a shifted dose–response since this affects both CFP and YFP equally, from true intrinsic stochasticity. (**B**) Overview of algorithm. By fitting a gamma distribution to the raw YFP data, calculating the CFP/YFP covariance and filtering this component out of the total variance, an intrinsic YFP distribution can be estimated (left). By repeating this for each dose–response distribution, intrinsic mutual information can be estimated (right). Full details on the algorithm are given in *Supplementary file 2*. (**C**) By applying the algorithm to the data from *Figure 2* (solid bars), we can estimate intrinsic mutual information (hatched bars).

The following figure supplement is available for figure 3:

**Figure supplement 1**. Input noise and variability in Msn2 abundance.

Third, keeping the mean constant, a new gamma distribution without the extrinsic component is inferred (YFP$_\text{int}$ ~ $\Gamma(a_\text{int}, b_\text{int})$). Fourth, this is repeated for each Msn2 input (e.g., amplitude or frequency). Finally, this inferred data set is discretized and then used to estimate $I_\text{int}$ (*Figure 3B*; see *Supplementary file 2* for a detailed discussion of the algorithm). We verified our algorithm in silico by systematically simulating five linear and five non-linear gene expression models with and without extrinsic noise and compared the true $I_\text{int}$ to the algorithm-inferred $I_\text{int}$. The algorithm tended to slightly underestimate the true $I_\text{int}$, but the mean error was less than 2% and the error was always less than 5% (*Supplementary file 2*).

Therefore, by using dual-reporter strains we can determine how much of the information loss is extrinsic, apply the algorithm and estimate $I_\text{int}$ in each case ($I_\text{AM,int}$ and $I_\text{FM,int}$). We find that filtering out extrinsic noise significantly increases $I$ (hatched bars, *Figure 3C*). Since the cell most likely incorporates some gene-extrinsic factors into a decision, but most likely does not incorporate all gene-extrinsic factors, we interpret $I_\text{raw}$ and $I_\text{int}$ as a lower and upper bound, respectively, on the true $I$. Thus, our approach allows us to estimate an upper bound, $I_\text{int}$, on a promoter's information transduction capacity.

Even after correcting for extrinsic noise, $I_\text{AM,int}$ for *HXK1* and *SIP18* only reach ~1.5–1.6 bits (*Figure 3C*). And $I_\text{FM,int}$ for *HXK1* is just 1.36 bits—that is, three ranges of inputs can only be distinguished with some associated error. Thus, even when considering $I_\text{int}$, which is the upper limit on the maximal mutual information, neither natural Msn2 target gene can transmit information about stress intensity without some error. That is, consistent with the 'noisy switch model', expression of *HXK1* and *SIP18* is not reliably fine-tuned to stress intensity. In contrast, for mut B, $I_\text{FM,int}$ is 1.55 bits and $I_\text{AM,int}$ is ~2 bits (*Figure 3C*). Thus, mut B almost approaches a range where information about both signal identity and intensity could conceivably be transduced nearly without error like a 'rheostat', though the natural Msn2 target genes, *HXK1* and *SIP18*, do not.

## Multiple gene copies reduce information loss due to intrinsic noise

Filtering out extrinsic noise substantially increases $I$ (*Figure 3C*). Next, we considered how reducing intrinsic noise might increase $I$. In principle, as the number of gene copies increases, information loss

due to intrinsic noise decreases due to simple ensemble averaging and mutual information increases—in the limit of infinite copies, intrinsic noise is zero and all information loss is due to extrinsic noise (*Cheong et al., 2011*). To test this we generated diploid strains with either one (1×) or two (2×) copies of the *hxk1*::CFP and *sip18*::YFP reporters in the same cell.

We repeated the AM and FM experiments for the 1× and 2× diploids (*Figure 4—figure supplement 1* and *Figure 4—figure supplement 2*). Comparing the 1× and 2× diploids (*Figure 4A*), we see that having two copies of a gene generally improves *I* by ~0.05–0.20 bits. For example, on going from haploid (1×) to diploid (2×), HXK1 $I_{AM}$ increases from 1.30 to 1.47 bits. Therefore, in terms of information transduction, being diploid confers a small but robust advantage.

## Circuits integrating the response of two genes can transduce more information than single gene circuits

So far we have considered information transduction from Msn2 to a single gene. Yet, Msn2 controls the expression of hundreds of genes in response to different stresses (*Elfving et al., 2014*; *Hao and O'Shea, 2012*; *Huebert et al., 2012*), we therefore extend our approach to information transduction from Msn2 to multiple genes. We next asked whether one way the cell might overcome the low *I* of individual genes would be to integrate the response of two or more different genes. To simulate and test this, we used diploid strains with both *hxk1*::CFP and *sip18*::YFP in the same cell, which allows us to measure the joint mutual information, $I(R_1,R_2;S)$.

We find that the AM joint mutual information ($I_{AM,joint}$) is significantly higher in both the 1× and 2× cases than the individual $I_{AM}$ of HXK1 and SIP18 (*Figure 4A*). For example, the total joint mutual information ($I_{AM+FM,joint}$; combining both the AM and FM responses) is 1.67 bits and 1.83 bits for the 1× and 2× diploids, respectively (*Figure 4A*). Therefore, although HXK1 and SIP18 individually can only distinguish ON from OFF without error (*Figure 2A*), their joint response can distinguish three inputs (no input, FM, or AM) nearly without error (*Figure 4B*).

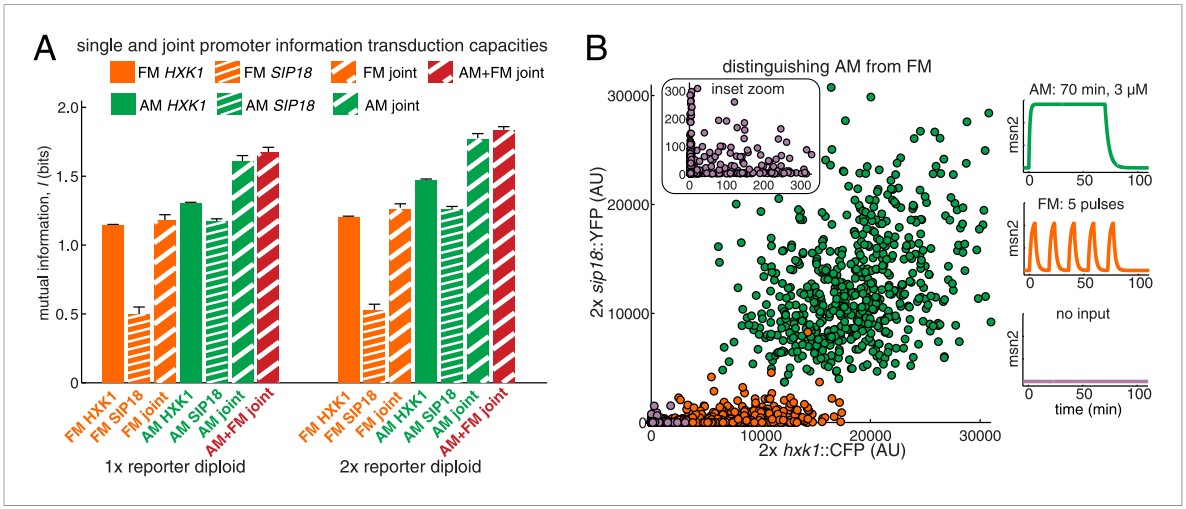

**Figure 4**. Integrating the response of more than one gene improves information transmission. (**A**) The AM and FM experiments (*Figure 2*) were repeated for diploid strains containing either one copy (1×) of the *hxk1*::CFP and *sip18*::YFP reporters or two copies (2×) of the *hxk1*::CFP and *sip18*::YFP reporters and individual and joint mutual information determined (full details on calculations are given in *Supplementary file 2*). (**B**) 2× *sip18*::YFP vs 2× *hxk1*::CFP scatterplot showing expression for three experiments: no input (light purple), five 5-min pulses of 690 nM 1-NM-PP1 separated by 13-min intervals (orange) or one 70-min pulse of 3 μM 1-NM-PP1 (green). For each condition, 600 cells are shown. The YFP/CFP expression is the maximal value after each time-trace has reached a plateau. The inset shows a zoom-in highlighting the 'no input' condition. All raw single-cell time-lapse microscopy source data for the 1× reporter diploid (21,236 cells) and 2× reporter diploid (19,222 cells) for this Figure are available online as *Supplementary file 1* and in (*Hansen and O'Shea, 2015*).

The following figure supplements are available for figure 4:

**Figure supplement 1**. Summary of results for 1× reporter diploid.

**Figure supplement 2**. Summary of results for 2× reporter diploid.

Thus, these results show that although the information transduction capacities of individual genes may be low, by integrating the response of two different genes the cell can improve information transduction. Therefore, by integrating the response of even more than two genes, the cell could potentially substantially improve the information transduction capacity of a pathway.

## Discussion

Here, we use information theory to investigate the hypothesis that cells can transduce both signal identity and signal intensity information in the amplitude and frequency of TF activation dynamics to control gene expression. As a conceptual framework, we introduce two extreme models of information transmission (*Figure 1C*): in the 'noisy switch model', the cell only transmits information sufficient to turn ON or OFF particular genes or pathways in response to external signals or stresses, whereas in the 'rheostat model' the cell is accurately fine-tuning the expression levels of relevant genes to the intensity of a signal or stress. For a TF responding to multiple stresses, we can extend this framework beyond a single gene. Extending the noisy switch model to two genes, the stress-relevant gene *HXK1* is reliably induced during FM pulsing of Msn2 (as seen under glucose limitation), whereas both *HXK1* and the stress-relevant *SIP18* gene are reliably induced during AM activation of Msn2 (as seen under oxidative stress) (*Figure 4B*). Therefore, three inputs (no input, FM, or AM) can be distinguished essentially without error (*Figure 4B*). However, given the modest joint information transduction capacities with respect to AM and FM combined ($I_{AM+FM,joint}$; *Figure 4A*), the cell cannot fine-tune *HXK1* and *SIP18* expression levels without significant error to the stress intensity. Thus, signal identity information for two distinct stresses can be transduced in the amplitude and frequency of Msn2 essentially without error, but intensity information can only be transduced with high error.

A central result in information theory is that the information transduction capacity of a signaling pathway is limited by and equal to the capacity of its weakest node or bottleneck (see also *Supplementary file 2* for a discussion). In other words, once information has been lost, no amount of post-processing can recover it, as is seen in the game of 'broken telephone'. Therefore, by measuring information transduction of individual Msn2 target genes to be ~1.0–1.3 bits, we can establish that the expression of Msn2 target genes cannot transduce stress signal intensity information without significant error at least for the AM and FM signals studied here—we can draw this conclusion without knowing all the relevant upstream components of the signaling pathway, how they mechanistically interact and how much information they can transmit. Thus, this approach can provide insight into the purpose of a pathway (e.g., noisy switch vs rheostat) and can readily be applied to other signaling pathways.

Why does information transduction by Msn2 resemble a 'noisy switch' rather than a 'rheostat'? Or phrased differently, why should the cell not fine-tune the expression level of stress genes to the stress intensity? One possibility is that the stochasticity inherent in the biophysical process of transcription fundamentally constrains information transduction by a promoter to ~1.0–1.3 bit. However, since the information transduction capacity of *SIP18* can be substantially increased by modulating promoter *cis*-elements (*Figures 2 and 3*), the low *I* of natural Msn2 target genes is not solely due to inherent biophysical constraints. Another speculative possibility is that variability is selected for: since evolutionary selection works at the population-level, variability in gene expression can create phenotypic diversity within an isogenic population (*Balaban et al., 2004*; *Blake et al., 2006*). It is also important to note that under natural stress a network of factors could be activated, whereas here we study the limits on amplitude- and frequency-mediated transduction of gene expression information in the dynamics of a single master TF.

Many biological signaling pathways transmit information through the amplitude or frequency of a shared signaling molecule (*Figure 1A*) and this has raised the long-standing question: can more information be transmitted through the amplitude or the frequency of a signaling molecule (*Rapp et al., 1981*; *Li and Goldbeter, 1989*)? This question has not previously been experimentally addressed for TFs responding to multiple signals in an amplitude- or frequency-dependent manner. We show that more gene expression information can be transduced through the amplitude than through the frequency of Msn2 activation dynamics for all genes studied here (*Figures 2 and 3*). Although the FM dose-responses tend to be more linear, the AM dose-responses have higher dynamic range and lower noise (*Figure 2* and *Figure 2—figure supplement 1D*). While we show that gene promoters have higher information transduction capacities for amplitude- than frequency-encoded information (*Figures 2 and 3*), maximal information transduction can be achieved for TFs that exhibit both amplitude- and frequency-encoding (*Figure 4*).

The amount of information promoters measured in this study can transmit is limited (*Figures 2–4*); yet we stress that for many 'house-keeping' genes or genes expressed at steady-state information transduction is likely significantly higher, in part due to time-averaging. Indeed, the gene expression response to a transient signal is noisier than a response at steady-state (*Hansen and O'Shea, 2013*) and inducible genes tend to show higher expression noise (*Bar-Even et al., 2006*; *Newman et al., 2006*). One way the cell can improve information transduction is by integrating the response of more than one gene or by having multiple copies of a gene (*Figure 4*). An example of this is ribosome biogenesis where, by having multiple copies of each gene encoding a subunit and employing elaborate feedback control, the cell can fine-tune its translational capability to its growth and energy status (*Lempiainen and Shore, 2009*). Another example is morphogen or cytokine secretion: although the amount produced by each single cell might be noisy, the average amount produced by a large number of cells can be highly precise (*Gregor et al., 2007*; *Cheong et al., 2011*). Hence, a number of strategies for increasing information transmission exist.

In conclusion, we have investigated the reliability of transmitting gene expression information in the amplitude and frequency of a TF. We show that the information transduction capacity of a gene can be tuned in *cis* and the amount of information transmitted increased by integrating the response of multiple genes. Nonetheless, for individual genes our results are consistent with the Msn2 pathway transmitting essentially error-free signal identity information, but unreliable signal intensity information, and therefore functioning more like a 'noisy switch' than a 'rheostat'. Since many similar master regulators, such as p53, NF-κB, ERK, and Hes1, also transduce information through the regulation of signaling dynamics, it will be interesting to investigate whether dynamic cell signaling is generally limited to error-free transduction of only signal identity information.

## Materials and methods

### Microfluidics and time-lapse microscopy

Microscopy experiments were performed essentially as described previously (*Hansen et al., 2015*; *Hansen and O'Shea, 2013*). Briefly, yeast cells were grown overnight at 30°C with shaking at 180 rpm to an $OD_{600 \text{ nm}}$ of ca. 0.1 in low fluorescence medium, quickly collected by suction filtration, loaded into the five channels of a microfluidic device pretreated with concanavalin A and the setup mounted on a Zeiss Axio Observer Z1 inverted fluorescence microscope (Carl Zeiss, Jena, Germany) equipped with an Evolve EM-CCD camera (Photometrics, Tuscon, AZ), 63× oil-immersion objective (NA 1.4, Plan-Apochromat), Zeiss Colibri LEDs for excitation and an incubation chamber kept at 30°C. Solenoid valves programmed using custom-written software (MATLAB) control whether medium with or without 1-NM-PP1 is delivered to each microfluidic channel and the flow (ca. 1 μL/s) is driven by gravity. Control of 1-NM-PP1 delivery enables the control of Msn2 pulsing (*Figure 2*) and a unique pulse sequence can be delivered to each of the five microfluidic channels. The microscope maintains focus and moves between each channel to acquire phase-contrast, YFP, CFP, RFP (mCherry), and iRFP images for 64 frames with a 2.5 min time resolution. For the AM experiments, 1-NM-PP1 was added to each microfluidic channel for 70 min at the following concentrations: 100 nM, 175 nM, 275 nM, 413 nM, 690 nM, 1117 nM, 3 μM. For the FM experiments, a concentration of 690 nM 1-NM-PP1 was used together with the following pulse sequences: one 5-min pulse; two 5-min pulses separated by a 40-min interval; three 5-min pulses separated by 25-min intervals; four 5-min pulses separated by 17.5-min intervals; five 5-min pulses separated by 13-min intervals; six 5-min pulses separated by 10-min intervals; seven 5-min pulses separated by 7.86-min intervals; eight 5-min pulses separated by 6.25-min intervals; nine 5-min pulses separated by 5-min intervals. Control software for the microfluidic device and a full protocol are provided elsewhere (*Hansen et al., 2015*). Image analysis was performed using custom-written software (MATLAB) that segments, tracks and quantifies single-cell time-traces and has been described previously (*Hansen et al., 2015*; *Hansen and O'Shea, 2013*). All raw single-cell data are available online as *Supplementary file 1* and in (*Hansen and O'Shea, 2015*).

### Computation of mutual information

The mutual information for a single reporter is defined in *Equation 1* and the maximal mutual information given by:

$$I(R; S) = \max_{p(S)}[MI(R; S)] \quad \text{for} \quad \sum_i p(S_i) = 1; \quad p(S_i) \geq 0.$$

The $p(S)$ that maximizes the mutual information is determined using the iterative Blahut-Arimoto algorithm. An unbiased $I$ was estimated using jackknife sampling to correct for undersampling as it has previously been described (**Strong et al., 1998**; **Slonim et al., 2005**; **Cheong et al., 2011**). The data were discretized by binning as shown in **Figure 2**. Maximal mutual information, $I$, and its error are reported as the mean and standard deviation, respectively, from calculating the unbiased $I$ using 15 to 35 bins, inclusive.

To determine the maximal joint mutual information, $I$ (**Figure 4A**), first consider the joint mutual information between the signal $S$ and two responses $R_1$ (e.g., YFP) and $R_2$ (e.g., CFP):

$$MI(R_1, R_2; S) = MI(R_1; S) + MI(R_2; S|R_1),$$

Where $MI(R_1;S)$ is known from **Equation 1** and $MI(R_2;S|R_1)$ is given by:

$$MI(R_2; S|R_1) = \sum_{i,j,k} p(R_1(i), R_2(j), S(k)) \log_2\left(\frac{p(R_1(i))p(R_1(i), R_2(j), S(k))}{p(R_1(i), R_2(j))p(R_1(i), S(k))}\right).$$

The maximal joint mutual information is then given by:

$$I(R_1, R_2; S) = \max_{p(S)}[MI(R_1, R_2; S)] \quad \text{for} \quad \sum_i p(S_i) = 1; \quad p(S_i) \geq 0.$$

As before, $p(S)$ is obtained using the Blahut-Arimoto algorithm, and the mean and error of $I$ are obtained as for a single reporter, except using 8 to 20 bins, inclusive. Full details are given in **Supplementary file 2**.

## Algorithm to estimate the intrinsic mutual information

Briefly, the total, intrinsic and extrinsic noise for each condition is calculated using dual-reporters (CFP/YFP) (**Elowitz et al., 2002**; **Swain et al., 2002**). The expression distributions in the absence of extrinsic noise are required to determine $I_{int}$. This is an intractable problem (**Hilfinger and Paulsson, 2011**). To estimate it, the raw, empirical YFP distribution is fitted to a gamma distribution (YFP $\sim \Gamma(a, b)$). Keeping the mean fixed, a new gamma distribution representing the YFP response in the absence of extrinsic noise is then inferred by filtering out the extrinsic contribution to the variance. This is repeated for each condition, each distribution is then discretized and the maximal mutual information, $I$, determined as above.

The accuracy of the algorithm was tested by simulating five linear and five non-linear stochastic gene expression models for both a fast and a slow promoter using the Gillespie algorithm under AM (10 conditions). Extrinsic noise is added by picking the translation rate and TF concentration for each iteration from a gamma distribution. The algorithm was then applied to each data set with extrinsic noise and compared to simulation results with only intrinsic noise and the error calculated. In all 80 cases (10 models, 2 promoters, 4 levels of extrinsic noise), the error was less than 5% (in bits) and the mean error was less than 2%. Full details are given in **Supplementary file 2**.

## Measurement noise, data processing, and YFP quantification

Measurement noise is a major concern for information theoretical calculations and can lead to underestimates of mutual information. To control and minimize effects of noise, the following data processing pipeline was employed. For each single-cell, a time-trace of 64 YFP measurements is made (2.5 min interval). The fluorescence (in AU) is the mean pixel-intensity per cell corresponding to the YFP concentration. As can be seen in **Figure 2—figure supplement 1B** and **Figure 2—figure supplement 2** from the single-cell YFP traces, YFP concentration generally reaches a plateau around or after the 100 min time-point (element 43 in the YFP vector). So the maximal YFP level in the cell is measured approximately 20 times before the experiment ends (element 64 in the YFP vector). Although there is slight noise in each measurement of the YFP concentration as shown in **Figure 2—figure supplement 2A** (black circles), because YFP is independently measured ~20 times after it has reached a plateau, the actual YFP level can accurately be determined by smoothing (**Figure 2—figure supplement 2A**, red line). The YFP trace is smoothed using an 11-point moving average filter and the vector is subsequently converted to a scalar by taking the maximal YFP value in the (33;64) range of elements. The scalar

YFP concentration (*Figure 2—figure supplement 1B*) is used for all information theoretical calculations. We believe that the protein concentration is the most biologically relevant measure of gene expression. For example, the activity of a stress response enzyme is generally determined by its concentration. But we note that had a different measure been used, that is, had the dynamics of the YFP time-trace been included, different estimates of $I$ would be obtained (see also *Supplementary file 2* for a further discussion).

The following factors, among others, contribute to measurement noise: slight variations in microscope focusing; fluctuations in cellular autofluorescence; instrumentation variability (e.g., camera noise); day-to-day experimental variability; slight errors from automated image analysis. Nonetheless, as is also evident from *Figure 2—figure supplement 2* measurement noise is small. For HXK1 and SIP18 $I_{AM}$ and $I_{FM}$ were independently measured twice in different strains: the *SIP18* dual-reporter strain (EY2813/ASH94), the *HXK1* dual-reporter strain (EY2810/ASH91), and the 1× reporter diploid (EY2972/ASH194). The results are shown in the table below:

| $I$ | *Gene*::YFP / *gene*::CFP strain | 1x *sip18*::YFP / *hxk1*::CFP strain |
| --- | --- | --- |
| $I_{AM}(sip18::YFP)$ | 1.21 ± 0.03 bits | 1.17 ± 0.02 bits |
| $I_{FM}(sip18::YFP)$ | 0.52 ± 0.06 bits | 0.50 ± 0.05 bits |
| $I_{AM}(hxk1::CFP/YFP)$ | 1.30 ± 0.01 bits | 1.30 ± 0.01 bits |
| $I_{FM}(hxk1::CFP/YFP)$ | 1.11 ± 0.01 bits | 1.14 ± 0.01 bits |

As is clear from the table above, the measurements of $I_{AM}$ and $I_{FM}$ between different strains (with slightly different genetic backgrounds) are highly similar and within error. This provides high confidence in the measurements and shows that the measurements are robust between different clones. Nonetheless, a constant noise source would cause all measurements to be underestimates by similar amounts. Therefore, the consistency of the measurements does not exclude the presence of a constant noise source. However, it is also important to note that most noise sources are 'extrinsic' to the gene and will therefore partially be filtered out by the algorithm during the correction for extrinsic noise.

## Strain construction

All strains used in this study are listed in *Table 1*. The diploid strains containing fluorescent reporters for the *SIP18* (ASH94/EY2813) and *HXK1* (ASH91/EY2810) promoters have been described previously (*Hansen and O'Shea, 2013*). These and all other *Saccharomyces cerevisiae* strains used in this study are from an $ADE^+$ strain in the W303 background (*MATa* [EY0690] and *MATα* (EY0691) *trp1 leu2 ura3 his3 can1 GAL$^+$ psi$^+$*). Standard methods for growing and genetically manipulating yeast were used throughout this study and all manipulations were performed in the same manner in both haploid mating types unless otherwise stated. Mating was performed by mixing haploids and selecting for diploids on SD–TRP–LEU plates. All genetic manipulations were verified by polymerase chain reaction (PCR).

To generate the *pSIP18* promoter mutants, the relevant segment of the promoter was replaced by *URA3* and followed by replacing the *URA3* fragment with a PCR generated fragment containing the relevant mutations and counterselection against *URA3*. The full sequence of the wild-type *SIP18* promoter and the mutant promoters is listed below.

### > WT *SIP18* promoter

GCTCACTTTTTGTTGGTCTGTATTCATTCTGGATGTCTTGGTTGTAGAAATTTCTTTTATTGGTTCATT
AAAGTCAAGGTAAATGGCGAGAACTAGAATAGAGTTTTATTCTTTTTACCGTTATATAGATAATTCT
AGCCGGGGGCGGTCG**CCCCT**GAGATTCCCGACATCAGTAAGACATAGTACTGTACGATTACTGT
ACGATTAATCTATCCACTTCAGATGTTCAACAATTCCTTTTGGCATTACGTATTAATACTTCATAGG
ATCGGCACCCTCCCTTAAGCCT**CCCCT**AAATGCTTTCGGTA**CCCCT**TTAAGACAACTATCTCTTAAC
CTTCTGTATTTACTTGCATGTTACGTTGAGTCTCATTGGAGGTTTGCATCATATGTTTAGGTTTTTTT
GGAAACGTGGACGGCTCATAGTGATTGGTAAATGGGAGTTACGAATAAACGTATCTTAAAGGGA
GCGGTATGTAAATGGATAGATGATCATGAATACAGTACGAGGTGTAAAGAATGATGGGACTGA
GAGGGCAATTATCATCCCTCAGAATCAACATCACAAACATATATAAAGCTCCCAATTCTGCCCCA
AAGTTTTGTCCCTAGGCATTTTTAATCTTTGTATCTGTGCTCTTTACTTTAGTAGAAAGGTATATAAA
AAAGTATAGTCAAG

**Table 1.** List of strains.

| Strain | Type | Strain details |
|---|---|---|
| EY0690 | MAT**a** | W303 (*trp1 leu2 ura3 his3 can1 GAL*+ *psi*+) (not generated in this study) |
| EY0691 | MATα | W303 (*trp1 leu2 ura3 his3 can1 GAL*+ *psi*+) (not generated in this study) |
| EY2808/ ASH89 | MAT**a** | *TPK1*M164G *TPK2*M147G *TPK3*M165G *msn4*Δ::*TRP1 MSN2*-mCherry *NHP6a*-iRFP::kanMX *hxk1*::mCitrine_V163A-*spHIS5* (not generated in this study) |
| EY2809/ ASH90 | MATα | *TPK1*M164G *TPK2*M147G *TPK3*M165G *msn4*Δ::*LEU2 MSN2*-mCherry *NHP6a*-iRFP::kanMX *hxk1*::SCFP3A-*spHIS5* (not generated in this study) |
| EY2810/ ASH91 | Diploid | *TPK1*M164G *TPK2*M147G *TPK3*M165G *msn4*Δ::*TRP1/LEU2 MSN2*-mCherry *NHP6a*-iRFP::kanMX *hxk1*::mCitrineV163A/SCFP3A-*spHIS5* (not generated in this study) |
| EY2811/ ASH92 | MAT**a** | *TPK1*M164G *TPK2*M147G *TPK3*M165G *msn4*Δ::*TRP1 MSN2*-mCherry *NHP6a*-iRFP::kanMX *sip18*::mCitrine_V163A-*spHIS5* (not generated in this study) |
| EY2812/ ASH93 | MATα | *TPK1*M164G *TPK2*M147G *TPK3*M165G *msn4*Δ::*LEU2 MSN2*-mCherry *NHP6a*-iRFP::kanMX *sip18*::SCFP3A-*spHIS5* (not generated in this study) |
| EY2813/ ASH94 | Diploid | *TPK1*M164G *TPK2*M147G *TPK3*M165G *msn4*Δ::*TRP1/LEU2 MSN2*-mCherry *NHP6a*-iRFP::kanMX *sip18*::mCitrineV163A/SCFP3A-*spHIS5* (not generated in this study) |
| EY2964/ ASH139 | MAT**a** | *TPK1*M164G *TPK2*M147G *TPK3*M165G *msn4*Δ::*TRP1 MSN2*-mCherry *NHP6a*-iRFP::kanMX *sip18*::mCitrine_V163A-*HIS3 pSIP18* Mut A 3 STREs |
| EY2965/ ASH140 | MAT**a** | *TPK1*M164G *TPK2*M147G *TPK3*M165G *msn4*Δ::*TRP1 MSN2*-mCherry *NHP6a*-iRFP::kanMX *sip18*::mCitrine_V163A-*HIS3 pSIP18* Mut B 4 STREs |
| EY2966/ ASH188 | MATα | *TPK1*M164G *TPK2*M147G *TPK3*M165G *msn4*Δ::*LEU2 MSN2*-mCherry *NHP6a*-iRFP::kanMX *sip18*::SCFP3A-*HIS3 pSIP18* Mut B 4 STREs |
| EY2967/ ASH189 | Diploid | *TPK1*M164G *TPK2*M147G *TPK3*M165G *msn4*Δ::*TRP1/LEU2 MSN2*-mCherry *NHP6a*-iRFP::kanMX *sip18*::mCitrine_V163A/SCFP3A-*HIS3 pSIP18* Mut B 4 STREs |
| EY2968/ ASH190 | MATα | *TPK1*M164G *TPK2*M147G *TPK3*M165G *msn4*Δ::*LEU2 MSN2*-mCherry *NHP6a*-iRFP::kanMX *sip18*::SCFP3A-*HIS3 pSIP18* Mut A 3 STREs |
| EY2969/ ASH191 | Diploid | *TPK1*M164G *TPK2*M147G *TPK3*M165G *msn4*Δ::*TRP1/LEU2 MSN2*-mCherry *NHP6a*-iRFP::kanMX *sip18*::mCitrine_V163A/SCFP3A-*HIS3 pSIP18* Mut A 3 STREs |
| EY2970/ ASH192 | MAT**a** | *TPK1*M164G *TPK2*M147G *TPK3*M165G *msn4*Δ::*TRP1 MSN2*-mCherry *NHP6a*-iRFP::kanMX *sip18*::mCitrine_V163A-*HIS3 hxk1*::*URA3* |
| EY2971/ ASH193 | MATα | *TPK1*M164G *TPK2*M147G *TPK3*M165G *msn4*Δ::*LEU2 MSN2*-mCherry *NHP6a*-iRFP::kanMX *hxk1*::SCFP3A-*HIS3 sip18*::*URA3* |
| EY2972/ ASH194 | Diploid | *TPK1*M164G *TPK2*M147G *TPK3*M165G *msn4*Δ::*TRP1/LEU2 MSN2*-mCherry *NHP6a*-iRFP::kanMX *sip18*::mCitrine_V163A-*HIS3 hxk1*::*URA3* / *hxk1*::SCFP3A-*HIS3 sip18*::*URA3* (1x reporter diploid) |
| EY2973/ ASH195 | MAT**a** | *TPK1*M164G *TPK2*M147G *TPK3*M165G *msn4*Δ::*TRP1 MSN2*-mCherry *NHP6a*-iRFP::kanMX *sip18*::mCitrine_V163A-*HIS3 hxk1*::SCFP3A-*HIS3* |
| EY2974/ ASH196 | MATα | *TPK1*M164G *TPK2*M147G *TPK3*M165G *msn4*Δ::*LEU2 MSN2*-mCherry *NHP6a*-iRFP::kanMX *hxk1*::SCFP3A-*HIS3 sip18*::mCitrine_V163A-*HIS3* |
| EY2975/ ASH197 | Diploid | *TPK1*M164G *TPK2*M147G *TPK3*M165G *msn4*Δ::*LEU2 MSN2*-mCherry *NHP6a*-iRFP::kanMX **2x** *hxk1*::SCFP3A_JCat-HIS3 **2x** *sip18*::mCitrine_V163A-*HIS3* (2x reporter diploid) |

> *pSIP18* mut A promoter

GCTCACTTTTTGTTGGTCTGTATTCATTCTGGATGTCTTGGTTGTAGAAATTTCTTTTATTGGTTCATT
AAAGTCAAGGTAAATGGCGAGAACTAGAATAGAGTTTTATTCTTTTTACCGTTATATAGATAATTCT
AGCCGGGGGCGGTCG**CCCCT**GAGATTCCCGACATCAGTAAGACATAGTACTGTACGATTACTGT
ACGATTAATCTATCCACTTCAGATGTTCAACAATTCCTTTTGGCATTACGTATTAATACTTCATAGG
ATCGGCACCCTCCCTTAAGCCT**CAACT**AAATGCTTTCGGTA**CAACT**TTAAGACAACTATCTCTTAA

CCTTCTGTATTTACTTGCATGTTACGTTGAGTCTCATTGGAGGTTTGCATCATATGTTTAGGTTTTTTT
GGAAACGTGGACGGCTCATAGTGATTGGTAAATGGGAGTTA**CCCCT**AAACGTATCTTAAAGGGA**C**
**CCCT**ATGTAAAATGGATAG**CCCCT**CATGAATACAGTACGAGGTGTAAAGAATGATGGGACTGAGA
GGGCAATTATCATCCCTCAGAATCAACATCACAAACATATATAAAGCTCCCAATTCTGCCCCAAA
GTTTTGTCCCTAGGCATTTTTAATCTTTGTATCTGTGCTCTTTACTTTAGTAGAAAGGTATATAAAAAA
GTATAGTCAAG

### > *pSIP18* mut B promoter

GCTCACTTTTTGTTGGTCTGTATTCATTCTGGATGTCTTGGTTGTAGAAATTTCTTTTATTGGTTCATT
AAAGTCAAGGTAAATGGCGAGAACTAGAATAGAGTTTTATTCTTTTTACCGTTATATAGATAATTCT
AGCCGGGGGCGGTCG**CCCCT**GAGATTCCCGACATCAGTAAGACATAGTACTGTACGATTACTGT
ACGATTAATCTATCCACTTCAGATGTTCAACAATTCCTTTTGGCATTACGTATTAATACTTCATAGG
ATCGGCACCCTCCCTTAAGCCT**CAACT**AAATGCTTTCGGTA**CAACT**TTAAGACAACTATCTCTTAA
CCTTCTGTATTTACTTGCATGTTACGTTGAGTCTCATTGGAGGTTTGCATCATATGTTTAGGTTTTTTT
TGGAAACGTGGACGGCTCATAGTGA**CCCCT**AAATGGGAGTTA**CCCCT**AAACGTATCTTAAAGGGA
**CCCCT**ATGTAAAATGGATAG**CCCCT**CATGAATACAGTACGAGGTGTAAAGAATGATGGGACTGAG
AGGGCAATTATCATCCCTCAGAATCAACATCACAAACATATATAAAGCTCCCAATTCTGCCCCAAA
GTTTTGTCCCTAGGCATTTTTAATCTTTGTATCTGTGCTCTTTACTTTAGTAGAAAGGTATATAAAAA
AGTATAGTCAAG

To remove the Msn2 binding site (STRE 5′-CCCCT′-3′), the two central Cs were replaced by As (5′-CCCCT′-3′ → 5′-CAACT′-3′), as shown in bold in the above sequences. The most upstream site in the *SIP18* promoter appears to be non-functional—deleting it has no effect on gene induction. Conversely, the two sites between −350 and −400 bp appear to be solely responsible for gene induction—deletion of both sites completely abolishes gene induction to below our detection limit. Mut A and Mut B have 3 and 4 new STRE sites, respectively, instead of the 2 STREs in the WT promoter. The position was chosen to be closer to the transcription start site, but in the largely nucleosome free region between two nucleosomes (*Figure 2—figure supplement 1C*). The same manipulations were performed in both mating types and all microscopy experiments were conducted in diploid strains (Mut A: EY2969/ASH191; Mut B: EY2967/ASH189).

To generate the 1× and 2× reporter diploid strains (1×: EY2972/ASH194; 2×: EY2975/ASH197), strain EY2811/ASH92 (MAT**a** *sip18*::mCitrineV163A-HIS) and strain EY2809/ASH90 (MATα *hxk1*:: SCFP3A-HIS) were used as base strains. In EY2811, the *HXK1* ORF was replaced by *URA3* to generate EY2970/ASH192, which was used for the 1× reporter diploid, and *URA3* further replaced by a PCR fragment containing SCFP3A followed by the *ADH1* terminator and the *spHIS5* selection marker (AddGene plasmid #64686) using counterselection against *URA3*. This gave strain EY2973/ASH195, which was used for the 2× reporter diploid. Likewise, in EY2811 the *SIP18* ORF was replaced by *URA3* to generate EY2971/ASH193, which was used for the 1× reporter diploid, and *URA3* further replaced by a PCR fragment containing mCitrineV163A followed by the *ADH1* terminator and the *spHIS5* selection marker (AddGene plasmid #64685) using counterselection against *URA3*. This gave strain EY2974/ASH196, which was used for the 2× reporter diploid. Furthermore, the 1× reporter diploid (EY2972/ASH194) was generated by mating EY2970/ASH192 and EY2971/ASH193 and the 2× reporter diploid (EY2975/ASH197) generated by mating EY2973/ASH195 and EY2974/ASH196. In the 1× reporter diploid, no WT copies of the *SIP18* and *HXK1* genes are present to ensure that, in the case the encoded protein product could have an autoregulatory effect, this complication would be avoided.

Finally, we note that 1-NM-PP1 mediated gene induction of *HXK1* and *SIP18* is specific to Msn2. In an *msn2Δ*-deletion strain, neither *HXK1* nor *SIP18* is induced by 1-NM-PP1 (*Hansen and O'Shea, 2013*) and both promoters have been shown to directly bind Msn2 in ChIP experiments (*Huebert et al., 2012*; *Elfving et al., 2014*).

All strains are available upon request and all strains are derived from EY0690 and EY0691.

## Acknowledgements

We thank Raymond Cheong, Gašper Tkačik, and Mikhail Tikhonov for insightful discussions. We thank Nan Hao, Dann Huh, Arvind Subramaniam, Matthew Brennan, Roshni Wadhwani, Andrian Gutu, Shankar Mukherji, Kapil Amarnath, Bodo Stern, Sharad Ramanathan and members of the O'Shea lab for discussions and critically reading the manuscript. This work

was performed in part at the Center for Nanoscale Systems at Harvard University, a member of the National Nanotechnology Infrastructure Network (NNIN), which is supported by the National Science Foundation under NSF award no. ECS-0335765. Image analysis and model simulations were run on the Odyssey cluster supported by the FAS Division of Science, Research Computing Group at Harvard University. The Howard Hughes Medical Institute supported this work.

## Additional information

### Competing interests

EKO'S, Chief Scientific Officer and a Vice President at the Howard Hughes Medical Institute, one of the three founding funders of *eLife*. The other author declare that no competing interests exist.

### Funding

| Funder | Grant reference | Author |
|---|---|---|
| Howard Hughes Medical Institute (HHMI) | | Erin K O'Shea |
| National Science Foundation (NSF) | ECS-0335765 | Anders S Hansen, Erin K O'Shea |

The funders had no role in study design, data collection and interpretation, or the decision to submit the work for publication.

### Author contributions

ASH, Conception and design, Acquisition of data, Analysis and interpretation of data, Drafting or revising the article, Contributed unpublished essential data or reagents; EKO'S, Conception and design, Drafting or revising the article

### Author ORCIDs

Anders S Hansen, http://orcid.org/0000-0001-7540-7858

## Additional files

### Supplementary files

• Supplementary file 1. Raw single-cell time-trace data for *HXK1* (15259 cells), *SIP18* (21242 cells), *pSIP18* mut A (18203 cells), *pSIP18* mut B (17655 cells), 1× reporter diploid (21236 cells), and 2× reporter diploid (19222 cells). The data are also available from Dryad Digital Repository (*Hansen and O'Shea, 2015*).

• Supplementary file 2. Computation of Mutual Information. Complete description of information theoretical computations and the algorithm.

### Major dataset

The following dataset was generated:

| Author(s) | Year | Dataset title | Dataset ID and/or URL | Database, license, and accessibility information |
|---|---|---|---|---|
| Hansen AS, O'Shea EK | 2015 | Data from: Limits on Information Transduction through Amplitude and Frequency Regulation of Transcription Factor Activity | http://dx.doi.org/10.5061/dryad.97vt8 | Available at Dryad Digital Repository under a CC0 Public Domain Dedication. |

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
