## [Decision Letter]

[Editors' note: this article was originally rejected after discussions between the reviewers, but the authors were invited to resubmit after an appeal against the decision.]

Thank you for choosing to send your work entitled “Limits on information transduction through regulation of signaling dynamics” for consideration at *eLife*. Your full submission has been evaluated by Detlef Weigel (Senior editor), a Reviewing editor and two peer reviewers. We are potentially very interested in the work—provided that your premises hold up—,but because it is *eLife* policy to only invite revision when it is clear that required experiments can be done in a short time frame, we are declining the work for now.

The reviewers were overall positive in the sense that it was agreed on all that the approach is interesting and that the results are potentially important, but the reviewers were concerned that you did not address the (likely) possibility of loss of information between input and MSN dynamics. If a large fraction of information is lost already at this point, this could, unfortunately, invalidate your claims.

Reviewer 2 suggests how this can be measured experimentally, and further specify a criterion to when this loss can be ignored (and when it cannot). We will be very interested in considering again a manuscript that includes such experiments. Since we are uncertain whether the results would indeed verify your model, we are declining the work for now, but we are leaving the door open for a re-submission, provided the new experimental data will be supportive of your model.

Reviewer #1:

The paper from Hansen et al. addresses a key open question in signal transduction on limits the ability of gene promoters to accurately to decode signaling dynamics. Overall the paper presents interesting results that I would like to see published in *eLife*. I do have a major concern about input noise that I am worried could invalidate many of the major results and therefore must be addressed. Furthermore, I found the writing of the paper to have many unnecessary overstatements and over simplistic interpretation that need to be corrected through a major revision of the text. I discuss two key concerns related to overstatements and over simplistic interpretation below. I made two suggestions that are constructive, but not essential. The first is for an additional analysis that I think could improve the paper and is easy to do and therefore I strongly recommend it will be performed. The second require additional experiment and while I think it would improve the paper substantially but might be outside the scope of this work and I do not see them as essential as long as the statements made by the authors are appropriately toned down.

Quantifying noise in inputs:

The system the authors analyze has three components: microfluidics manipulation => MSN2 translocation => gene expression. They analyze the mutual information between microfluidics input class and gene expression. Therefore they make an implicit assumption that the information transmission between microfluidics class and the patterns of MSN-2 is “noise free” and that all the information loss is in the decoding MSN2=>YFP. However, in their 2013 MSB paper, the authors showed non-negligible variability in MSN-2-mCherry localization within their microfluidics setup. Furthermore, other factors could contribute to this such as the variation in MSN2-mCherry between cells. There are two ways to address this issue. The best way will be to repeat the experiments when measuring both MSN-2-mCherry localization and gene expression in the same cells and to calculate the MI between all steps in the pathway similar to [70]. However, this might require substantial new experiments that are potentially beyond the scope of this work. The second best way is to just show the mutual information between the inputs and MSN-2 dynamics (in absolute units to address MSN2-mCherry concentration variability issues). If this value is close to 3 (log2(8)) than the assumption that the input noise is negligible is justified is reasonable, otherwise the information loss could be in the “signaling” or the response . They should have at least some of the required for this in their MSB 2013 paper. If there is substantial information loss between microfluidics and MSN2 response then I must recommend that the paper be rejected and more experiment done to carefully analyze loss at different steps along the artificial “pathway”.

Statements that need to be revised:

Real upper bound limits on dynamics?

The claim that this paper shows a limit on decoding of dynamics is overstated. The real physiological dynamics of MSN-2 are much more complex than simple amplitude and frequency as shown by Nan and O'Shea, Nat Struct Mol Biol, 2012. It is very likely that the *SIP18* and *HXK1* are tuned to the real physiological dynamics and not to artificial AM and FM signals. In fact the authors actually show in Figure 2 that these promoters are not optimized for these simple modes! The authors should restrict their claims to the information transduction through AM and FM signaling. This needs to be revised throughout the paper, Title, Abstract, main text etc.

Number of distinguishable states interpretation.

In this paper the authors constantly interpret their results as the number of distinguishable states. While this simplistic interpretation is tempting, I find it to be misleading for reasons explained nicely in Bowsher et al. 2014 which the authors cite. Mutual information should be interpreted as the increase in actionable information the cells has. Even a value of 0.77 bit could allow distinguishing between three states with some small associated inference error (see Bowsher 2014 Figure 1 and discussion in Box 2). I found section 2.4 in [Supplementary-material SD2-data] that was supposed to address this issue to be very lacking. The paper should be revised completely to address this point including removal of Figure 5 and the many statements that argue that cells are limited to decoding identity and not intensity.

The use of information theory analysis is a great tool to analyze noise in signal transduction networks. This is a complicated tool and needs to be interpreted with care. However, as a community there was substantial disservice to this approach by using the over-simplistic interpretation of bits as number of distinguishable states. This was done in previous works by others and resulted in unnecessary resentment to information theory approaches. While I understand that it is a bit more challenging to write a paper that uses the more complex and accurate interpretation of the bit value, it is essential that we do so.

Constructive suggestions:

1) Compare analysis from Figure 3 to Figure 4.

It would be interesting to see in Figure 4 additional bars that show the mutual information done on the CFP/YFP cells from Figure 3. I believe that an analysis of the joint response of I(CFP, YFP; input) where CFP and YFP should show higher mutual information than a simple diploid YFP cell and basically similar level to the intrinsic mutual information calculated by the authors. This will provide validation to the analysis shown in Figure 3 and will allow better comparison of the results of the diploid.

2) Analyze the mutant promoters from Figure 2 in respect to physiological inputs.

Experiments that could be very helpful in general in addressing some of the issues mentioned above are the calculation of mutual information on physiological responses and not just manipulation of the dynamics. Specifically, it would be great to see if the mutants from Figure 2 also increase the mutual information from physiological response? I suspect that they will not and this will provide a relatively easy way to show that information is really encoded and decoded in the complex pattern of MSN-2 dynamics that goes beyond AM and FM.

Reviewer #2:

In this manuscript, Hansen and O'Shea reported the information theory-based analysis of the Msn2 signal transduction system. By controlling Msn2-mCherry nuclear localization dynamics and measuring its downstream target promoters, the authors revealed that the extent of information transduction for a single target promoter depends on the dynamics of Msn2 (AM or FM) as well as the number of STRE sites on the promoter. Additionally, the authors showed that integration of multiple target promoters enhances information transduction. Overall, I really like the information-theoretical analysis of signal transduction. While the conclusions presented are not overly exciting, i.e. there are limits on information processing through Msn2 dynamics, the experiments and analysis presented in this work opens up a new and interesting way of understanding single-cell signaling dynamics in general.

Major comments:

1) Cellular signal transduction is composed two steps: encoding step where (chemical) inputs are encoded into intracellular representation such as Msn2 dynamics, and decoding step where Msn2 dynamics are then decoded into target expression. In this manuscript, the authors focused on the decoding step, i.e., from Msn2 dynamics to promoter output. In all the analysis, the authors have assumed that Msn2 dynamics always follows the input chemical signal in every cell and thus the input TF signal equates the external chemical waveforms. However, I feel this assumption could present a fundamental problem in the analysis since it is hard to imagine that every cell in the population has the same Msn2 dynamics under a defined chemical waveform. More specifically, the distribution of responses (YFP level) could likely arise from the heterogeneous Msn2 dynamics among a population. Therefore, I feel that uncoupling the heterogeneity of Msn2 dynamics seems necessary for understanding how Msn2 dynamics contribute to the extent of information transduction.

2) Following the point above, the observed difference in signal transmission capacity between AM and FM Msn2 input may likely due to the difference in the degree of heterogeneity of Msn2 dynamics. For example, Msn2 may not follow the FM chemical signal as faithfully as the AM chemical signal. Thus, Msn2 dynamics could be more heterogeneous among cells in FM condition than in AM condition, leading to a more faithful signal transmission for AM. One might need to compare the variability in Msn2 response between AM and FM in order to study the role of Msn2 dynamics in the extent of information transmission in these conditions.

3) It occurred to me that the extent of information transduction positively correlates with the dynamic range of promoter response (Figure 2). In other words, by simply looking at the top rows (i.e., dose response curves) of each figure panel in Figure 2, I can immediately tell which condition transmits the most information (i.e., mut B AM). Is there any underlying principle that results in such a correlation? This correlation suggests that the dynamic range of the measurement may somehow affect the calculation of mutual information. A potential way to test if this is the case is to characterize the mutual information of the same condition under different lamp power or camera gain settings.

4) Regarding the calculation of the mutual information, maximum YFP level was used. The authors made the argument that final protein level is a biologically relevant quantity, which I agreed. In section 2.5 of [Supplementary-material SD2-data], the authors made arguments about why other quantities are less desirable. I think the authors may want to make the argument more quantitatively. It could be that maximum YFP is the least noisy quantity and thus most suited for calculating mutual information. Such argument can be supported by comparing the CV of possible quantities, such as rate of YFP production, max YFP level, YFP level at a chosen time, etc..

5) I am not sure if the authors performed control experiments (in this or previous papers) to show that all the YFP expression (from promoters studied) comes from Msn2 alone (i.e., no other regulators involved). In other words, deletion of Msn2 abolishes the promoter expression under 1-NM-PP1.

[Editors' note: what now follows is the decision letter after the authors submitted for further consideration.]

Thank you for choosing to send your work entitled “Limits on information transduction through regulation of signaling dynamics” for consideration at *eLife*. Your article and your letter of appeal have been considered by the original reviewers of the manuscript, and there are still some issues that need to be resolved before we can accept a revised paper.

Specifically, the appeal letter now includes the new experiment suggested by the reviewers, in which the information loss between the input and MSN2 dynamics is measured. As the reviewers originally worried, this information loss is quite substantial. This is a bit worrisome, especially as this value is inconsistent with lower loss inferred from the instrinsic/extrinsic analysis (Figure 3). Another issue is that this loss was measured only for the AM signal, and not from the FM signal. This may call for a major rethinking of how to interpret the results in the paper, and in this context, the reviewers suggested the following:

1) Change some of the interpretation of their data and present the paper with a careful analysis of the information loss due to intrinsic and extrinsic noise sources. This way, the fact that there is substantial loss between chemical input and MSN-dynamics is not a problem anymore, rather an interesting result. Between that and the comparison of AM/FM and the different mutants with increased dynamic range, there should be enough there for an interesting paper. The inconsistencies of the two methods would need to be addressed of course.

2) Quantify more directly the information transmission capacity between MSN2 dynamics and MSN2 promoter by measuring in the same cell MSN2 the dynamics of localization in the nucleus and the resulting promoter reporter. Since mutual information is a symmetric quantity, one could bin the promoter response into 8 or 16 bins. Than calculate the mutual information between the scalar MSN2 promoter response and the distribution of multivariate dynamics responses in each bin. This could be done by pooling all the chemical input data together and use an approach such as described e.g. in [58] to calculate mutual information between scalar input and dynamic response. Perhaps the data in Figure 2—figure supplement 2 is sufficient, or if not, additional experiments are required.

---

## [Author Response]

[Editors' note: the author responses to the first round of peer review follow.]

Thank you for reviewing our submission “Limits on information transduction through regulation of signaling dynamics”. In your decision you state that you are interested in a revised manuscript provided that our premises hold up, but you decline our manuscript for now because it is not clear whether the suggested experiments can be performed in a short timeframe. You state that: “We will be very interested in considering again a manuscript that includes such experiments”.

The major concern is whether variability in Msn2 dynamics for a given microfluidic 1-NM-PP1 input invalidates our claims. The reviewers are very clear: they suggest a specific experiment to measure mutual information between Msn2 nuclear localization and 1-NM-PP1 input. We performed this experiment (please see below).

However, we would like to stress that we can also address the major concern of the reviewers with the algorithm in Figure 3 of the manuscript. Even if there is high cell-to-cell variability in Msn2 dynamics in response to a specific 1-NM-PP1 input (extrinsic noise), this will affect the dual CFP and YFP expression reporters in the same cell to the same extent. When we apply our algorithm, we filter out extrinsic noise irrespective of its magnitude. Thus, when we calculate *I*_int_, we estimate what *I* would have been, had there been no variability in Msn2 dynamics.

We apologize if the algorithm and its use were not clear in our original submission. But to understand why, consider for example a cell with a very high concentration of Msn2. In this cell, both the CFP and YFP gene expression reporter will tend to show high expression. By measuring the co-variance between CFP and YFP in the same cell across the cell population, we can infer the extent to which their shared environment contributes to the observed gene expression variability (extrinsic noise). This shared environment includes variability in Msn2 level, Msn2 dynamics in response to 1-NM-PP1 input, cell cycle phase or any other variable that affect CFP and YFP levels in a correlated manner. Using the measured CFP/YFP co-variance and our algorithm, we can determine the intrinsic mutual information (*I*_int_ in Figure 3). *I*_int_ is the information transduction capacity of a promoter in the absence of extrinsic noise, that is, *I*_int_ is what *I* would have been had there been no Msn2 variability or input noise. Thus, *I*_int_ is the quantity the reviewers are asking for.

The reviewers made insightful suggestions regarding improvement of the text and further analysis of existing data. Since we can address the major concern of the reviewers—Msn2 input noise—with our algorithm (Figure 3), we would like to submit a revision of our manuscript that incorporates these suggestions.

In this letter, we first discuss the experiment the reviewers requested in detail. Next, we elaborate on our algorithm (Figure 3) and illustrate how this algorithm actually addresses the issue of Msn2 input noise independently of the result of the experiment requested by the reviewers.

*Quantifying input noise between 1-NM-PP1 and Msn2*:

We thank the reviewers and the editor for carefully reviewing our manuscript and emphasizing the crucial point regarding Msn2 input noise, and also for being very clear about which experiment they would like to see performed. Specifically, they want us to “… show the mutual information between the inputs and MSN-2 dynamics (in absolute units to address MSN2-mCherry concentration variability issues). If this value is close to 3 (log2(8)) than the assumption that the input noise is negligible is justified is reasonable… ”.

We performed this experiment. We exposed cells to a 70-min pulse of either no 1-NM-PP1 or one of the seven 1-NM-PP1 concentrations used in Figure 2 of the manuscript. We measured cell-to-cell variability in nuclear Msn2-mCherry using time-lapse microscopy. There are three major possible sources of Msn2 variability:

a) Measurement noise when quantifying how much Msn2-mCherry is nuclear.

b) Input noise when microfluidic 1-NM-PP1 input is converted into Msn2 dynamics.

c) Biological variability in Msn2 expression between cells.

Regarding measurement noise (a), we measured Msn2-mCherry nuclear localization every 10 min to minimize photobleaching. A major technical challenge is that the nucleus moves in and out of focus in diploid yeast cells during time-lapse experiments. We therefore collected a z-stack series of images for each time point. We quantified Msn2 nuclear localization in absolute units as the average nuclear level during the 70-min pulse. This also corresponds to the Msn2 AUC. The result is shown in Figure 3—figure supplement 1. However, Msn2 is a low abundance protein present at only a few hundred molecules per cell. Therefore, to measure the difference between 25% and 37.5% of maximal nuclear localization (100 nM vs. 175 nM 1-NM-PP1), our measurement precision has to be on the order of tens of molecules. This is challenging even with state-of-the-art equipment. Therefore, our measurements likely slightly overestimate variability in nuclear Msn2-mCherry.

We cannot readily distinguish (b), input noise, from measurement noise. However, when we expose cells to 1-NM-PP1 in the microfluidic chemostat all cells in a field of view respond deterministically by activating Msn2 with essentially identical kinetics (see Figure 3—figure supplement 1). Therefore, input noise when 1-NM-PP1 is converted into Msn2 dynamics is likely relatively minor. More importantly, however, our algorithm allows us to correct for variability in Msn2 dynamics regardless of the origin and magnitude of the variability.

Based on our measurements (Figure 3—figure supplement 1), the mutual information between the 1-NM-PP1 input and Msn2 nuclear concentration is ∼2.06 bits. This is lower than the maximal value of 3 bits. However, a value of 3 bits would mean that there was no cell-to-cell variability in the Msn2 concentration. We estimate that the variability in nuclear Msn2-mCherry between cells is less than CV∼15% (σ/μ or std/mean; see Figure 3—figure supplement 1). This is an upper bound because any measurement noise would cause us to overestimate the Msn2 abundance CV. To put this into context, a proteomic study measured cell-to-cell variability among ∼2,500 yeast proteins (50). According to this study, essentially no yeast proteins have a CV below 10% (Figure 2 in [50]).

How low would the cell-to-cell variability of Msn2 have to be in order to obtain 3 bits? In other words, is it biologically possible to obtain a value of 3 bits? To investigate this, we simulated Msn2 variability and calculated mutual information for different CV's assuming that Msn2 abundance is gamma distributed. We simulated cells within a range of CV=20% to CV=10%, which is the lowest level of variability observed for any yeast protein (50). We find that if Msn2 was one of the least variable proteins in yeast with a CV∼10%, we would still expect mutual information less than 2.4 bits even if there was no loss of information between input and Msn2 dynamics (Figure 5). Furthermore, we emphasize that biological variability in Msn2 abundance between cells is not an artifact of our experimental system, but an inherent property of any biological pathway: a population of yeast cells responding to natural stress will have similar levels of variability in Msn2 abundance between cells.

Author response image 1.Msn2 nuclear localization was simulated with different CVs of Msn2 abundance. Briefly, we assume that Msn2 abundance is gamma distributed and that a given percentage of Msn2 is nuclear for a given inhibitor concentration (e.g. 50% for 275 nM or 75% for 690 nM in accordance with our measurements). The mean Msn2 abundance in the cell was chosen to be 1000 AU. A gamma distribution was then simulated with *a* and *b* parameters chosen such that the indicated CV was obtained. Note that the simulation with CV=15% (2.04 bits) closely matches our experimental result (2.06 bits).**DOI:**
http://dx.doi.org/10.7554/eLife.06559.018

We would also like to address Reviewer 2's specific concern regarding FM input. Reviewer 2 wrote: “for example, Msn2 may not follow the FM chemical signal as faithfully as the AM chemical signal”. During all experiments for this manuscript we simultaneously measured Msn2-mCherry and CFP and YFP gene expression in the same cell. We did not do a finely spaced z-stack series, which is necessary to accurately quantify the concentration of Msn2 in the nucleus—this causes too much photobleaching to be compatible with imaging at high temporal resolution. However, as shown in Figure 3—figure supplement 1 and in our previous paper (30), Msn2-mCherry deterministically and faithfully follows the 1-NM-PP1 input during FM as well as AM. Finally, even if there had been higher input noise during FM, this would affect the CFP and YFP gene expression reporters in a correlated manner. Therefore, we correct for this when we apply our algorithm and calculate *I*_int_.

*An algorithm to filter out extrinsic noise*:

The reviewers have pointed out the issue of noise in Msn2 dynamics—either coming from variability in Msn2 abundance between cells or from variability in how 1-NM-PP1 input is converted into Msn2 dynamics—which would cause us to underestimate *I*. There are several other extrinsic variables that will have the same effect. For example, previous studies have shown that the rate of transcription varies approximately 2-fold across the cell cycle in yeast (78). Thus, since we do not synchronize cells, some of the gene expression variability we observe is really because we do not control for cell cycle phase. However, the advantage of our approach is that we can correct for variability in Msn2 abundance, cell cycle phase and all other extrinsic variables, without specifying them or knowing which are the most important.

Since this is a crucial point, we briefly discuss the algorithm here (a full discussion is given in Section 4 of [Supplementary-material SD2-data]). We take advantage of a key insight from the gene expression noise field, where noise is divided into intrinsic and extrinsic noise (24). Intrinsic noise is due to the inherent stochasticity of biochemical reactions. Extrinsic noise, the extent to which the shared intracellular environment contributes to cell-to-cell variability, can be measured using two equivalent gene expression reporters (e.g. CFP and YFP). This is because the intracellular environment affects the CFP/YFP reporters to the same extent. The relative contributions of intrinsic and extrinsic noise are shown in Figure 2—figure supplement 1.

Suppose for example that gene expression variability was mainly caused by differences in Msn2 abundance between cells. In a cell with high Msn2 concentration, both CFP and YFP expression would be high. Conversely, in a cell with low Msn2 concentration, both CFP and YFP expression would be low. In this example, the CFP-YFP co-variance would be very high. Our algorithm allows us to use the CFP-YFP co-variance to estimate how much of the information loss is due to extrinsic noise (e.g. variability in Msn2 abundance) and, crucially, to calculate what *I* would have been in cell without extrinsic noise. As mentioned, there are many extrinsic variables such as Msn2 levels, cell cycle, cell size etc. Although it is in principle possible to correct for each extrinsic variable individually, our algorithm allows us to correct for all extrinsic variables at once and therefore to estimate an upper bound on information transduction of a promoter, *I*_int_.

Reviewer #1:

*The paper from Hansen et al. addresses a key open question in signal transduction on limits the ability of gene promoters to accurately to decode signaling dynamics. Overall the paper presents interesting results that I would like to see published in* eLife*. I do have a major concern about input noise that I am worried could invalidate many of the major results and therefore must be addressed. Furthermore, I found the writing of the paper to have many unnecessary overstatements and over simplistic interpretation that need to be corrected through a major revision of the text. I discuss two key concerns related to overstatements and over simplistic interpretation below. I made two suggestions that are constructive, but not essential. The first is for an additional analysis that I think could improve the paper and is easy to do and therefore I strongly recommend it will be performed. The second require additional experiment and while I think it would improve the paper substantially but might be outside the scope of this work and I do not see them as essential as long as the statements made by the authors are appropriately toned down*.

We thank the reviewer for their fair and constructive comments and suggestions. Regarding the first suggestion made by Reviewer 1, we have performed the additional analysis suggested by Reviewer 1. To address the second major point of Reviewer 1, as described below, we have toned down and re-written the relevant sections and claims.

Quantifying noise in inputs:

*The system the authors analyze has three components: microfluidics manipulation => MSN2 translocation => gene expression. They analyze the mutual information between microfluidics input class and gene expression. Therefore they make an implicit assumption that the information transmission between microfluidics class and the patterns of MSN-2 is “noise free” and that all the information loss is in the decoding MSN2=>YFP. However, in their 2013 MSB paper, the authors showed non-negligible variability in MSN-2-mCherry localization within their microfluidics setup. Furthermore, other factors could contribute to this such as the variation in MSN2-mCherry between cells. There are two ways to address this issue. The best way will be to repeat the experiments when measuring both MSN-2-mCherry localization and gene expression in the same cells and to calculate the MI between all steps in the pathway similar to*
[70]*. However, this might require substantial new experiments that are potentially beyond the scope of this work. The second best way is to just show the mutual information between the inputs and MSN-2 dynamics (in absolute units to address MSN2-mCherry concentration variability issues). If this value is close to 3 (log2(8)) than the assumption that the input noise is negligible is justified is reasonable, otherwise the information loss could be in the “signaling” or the response . They should have at least some of the required for this in their MSB 2013 paper. If there is substantial information loss between microfluidics and MSN2 response then I must recommend that the paper be rejected and more experiment done to carefully analyze loss at different steps along the artificial “pathway”*.

We performed the experiment that Reviewer 1 recommended and measured the mutual information between 1-NM-PP1 input and nuclear Msn2-mCherry for both AM and FM and quantified information loss. We estimate that *I*_AM_(Msn2; 1-NM-PP1)≥2.06 bits and *I*_FM_(Msn2; 1-NM-PP1)≥2.23 bits, but note that these are challenging measurements subject to measurement noise. The values given are therefore likely underestimates. We show the results in the new Figure 3—figure supplement 1.

Most importantly, we correct for input noise (1-NM-PP1→Msn2) and variability in Msn2 abundance with our algorithm in Figure 3. As Reviewer 1 also states, the key goal is estimating *I*(YFP; Msn2). As illustrated in Figure 6, we can estimate an upper limit on *I*(YFP; Msn2). As both reviewers pointed out, *I*(YFP; 1-NM-PP1) from Figure 2 is a lower bound on *I*(YFP; Msn2) since there is some information loss from 1-NM-PP1→Msn2 due to cell-to-cell variability in Msn2 abundance. Our algorithm allows us to estimate *I*_int_(YFP; 1-NM-PP1), which is an upper limit on *I*(YFP; Msn2), because any information loss from 1-NM-PP1→Msn2 will affect the dual CFP/YFP reporters in a correlated manner and we can correct for this information loss when we filter out extrinsic noise (Figure 3). Therefore, we can estimate both the lower (Figure 2) and upper (Figure 3) limit on *I*(YFP; Msn2) and thus achieve bounds on *I*(YFP; Msn2). Since *I*(YFP; Msn2) is the quantity both we and the reviewers are interested in, we believe our algorithm addresses this concern. We have re-written the manuscript text pertaining to Figure 3 to make clear how the algorithm corrects for Msn2 input noise.

Author response image 2.**DOI:**
http://dx.doi.org/10.7554/eLife.06559.019

*Statements that need to be revised*:

Real upper bound limits on dynamics?

*The claim that this paper shows a limit on decoding of dynamics is overstated. The real physiological dynamics of MSN-2 are much more complex than simple amplitude and frequency as shown by Nan and O'Shea, Nat Struct Mol Biol, 2012. It is very likely that the* SIP18 *and* HXK1 *are tuned to the real physiological dynamics and not to artificial AM and FM signals. In fact the authors actually show in*
Figure 2
*that these promoters are not optimized for these simple modes! The authors should restrict their claims to the information transduction through AM and FM signaling. This needs to be revised throughout the paper, Title, Abstract, main text etc*.

Studying the response to natural stress is an interesting future direction. Previously, we used “dynamics” as shorthand for AM/FM signaling. Following the recommendation of Reviewer 1, we now explicitly say AM/FM signaling instead of “dynamics” throughout the text, Title and Abstract.

*Number of distinguishable states interpretation*.

*In this paper the authors constantly interpret their results as the number of distinguishable states. While this simplistic interpretation is tempting, I find it to be misleading for reasons explained nicely in Bowsher et al. 2014 which the authors cite. Mutual information should be interpreted as the increase in actionable information the cells has. Even a value of 0.77 bit could allow distinguishing between three states with some small associated inference error (see Bowsher 2014*
Figure 1
*and discussion in Box 2). I found section 2.4 in*
[Supplementary-material SD2-data]
*that was supposed to address this issue to be very lacking. The paper should be revised completely to address this point including removal of Figure 5 and the many statements that argue that cells are limited to decoding identity and not intensity.*

*The use of information theory analysis is a great tool to analyze noise in signal transduction networks. This is a complicated tool and needs to be interpreted with care. However, as a community there was substantial disservice to this approach by using the over-simplistic interpretation of bits as number of distinguishable states. This was done in previous works by others and resulted in unnecessary resentment to information theory approaches. While I understand that it is a bit more challenging to write a paper that uses the more complex and accurate interpretation of the bit value, it is essential that we do so*.

This is a crucial point and we thank Reviewer 1 for their comments. Following Reviewer 1's comments, we have completely revised all discussions of “bits” and we have systematically removed interpretations such as “binary/ternary decision”. We have removed all “number of distinguishable states” annotation from the figures. And we have completely re-written and greatly expanded the discussion in section 2.4 [Supplementary-material SD2-data] using the Bowsher and Swain example Reviewer 1 refers to (12).

As Reviewer 1 points out, *I* = 1 bit may allow error free distinction between two states but it also may not. Likewise, *I* < 1 bit may allow distinguishing between multiple states with some error. Therefore, it is important to also inspect the probability distributions when interpreting *I*. Considering the distributions in Figure 2, we see that both *HXK1* and *SIP18* can distinguish ON (‘brown distribution’) from OFF without error. Based on Figure 2, we believe that this is clear. This is what we meant in our statements about “reliable transduction about signal identity, but not signal intensity information”. But as Reviewer 1 points out, it is correct that some information about signal intensity is also transmitted, albeit with a seemingly high error rate based on Figure 2. Therefore, we believe that our results are consistent with essentially error-free transduction of signal identity information, but that transduction of signal intensity information is only possible with some associated error. As Reviewer 1 points out, in some instances this associated error could be small. We thank Reviewer 1 for pointing this out and we have attempted to make this distinction much more clearly in the text and by including the new Figure 4.

*Constructive suggestions*:

*1) Compare analysis from*
Figure 3
*to*
Figure 4.

*It would be interesting to see in*
Figure 4
*additional bars that show the mutual information done on the CFP/YFP cells from*
Figure 3*. I believe that an analysis of the joint response of I(CFP, YFP; input) where CFP and YFP should show higher mutual information than a simple diploid YFP cell and basically similar level to the intrinsic mutual information calculated by the authors. This will provide validation to the analysis shown in*
Figure 3
*and will allow better comparison of the results of the diploid*.

We thank the reviewer for the suggestion and have performed this analysis. When we refer to “intrinsic” and “extrinsic” noise in this paper, we follow the “gene-centric” definition by Elowitz et al. (24). In principle, in the limit of infinite promoter copies, there should be no intrinsic noise due to ensemble averaging and all information loss comes from extrinsic noise. Conversely, when we filter out extrinsic noise, we can determine what *I* is when there is no information loss from gene-extrinsic sources, but all information loss comes from gene-intrinsic noise. Therefore, the joint *I*(CFP,YFP) corresponds to reducing intrinsic noise by adding one extra promoter copy, which is the same as what we did in Figure 4 when we compare the 1x and 2x reporter diploids. Thus, the joint *I*(CFP,YFP) can provide an independent confirmation of the 1x vs. 2x reporter comparison in Figure 4.

Using our dual YFP/CFP reporter data for *SIP18*, *HXK1*, mut A and mut B, we calculated *I*_AM,joint_(CFP,YFP) and *I*_FM,joint_(CFP,YFP) and now show this result in [Supplementary-material SD2-data] section 3.3..

In general, since the effect of adding an additional promoter leads to only a small increase in *I* the effect is difficult to measure precisely given the error in our mutual information estimates. Nonetheless, within error, we see agreement between the 1x/2x comparisons.

*2) Analyze the mutant promoters from*
Figure 2
*in respect to physiological inputs*.

*Experiments that could be very helpful in general in addressing some of the issues mentioned above are the calculation of mutual information on physiological responses and not just manipulation of the dynamics. Specifically, it would be great to see if the mutants from*
Figure 2
*also increase the mutual information from physiological response? I suspect that they will not and this will provide a relatively easy way to show that information is really encoded and decoded in the complex pattern of MSN-2 dynamics that goes beyond AM and FM*.

We agree that studying information transduction between stress and Msn2 and also between Msn2 and gene expression (Stress → Msn2 → Gene expression, e.g. using the approach of Uda et al. (70)) is a very interesting future direction, but this is beyond the scope of this work. We would like to emphasize that the natural dynamics of Msn2 under stress are noisy (33). One advantage of our synthetic 1-NM-PP1 approach is that we can more precisely control Msn2 translocation dynamics such as pulse number or frequency and therefore more accurately measure *I* for promoters. However, we agree with Reviewer 1 that the promoter response to natural dynamics could be different, but is beyond the scope of this study.

Reviewer #2:

*1) Cellular signal transduction is composed two steps: encoding step where (chemical) inputs are encoded into intracellular representation such as Msn2 dynamics, and decoding step where Msn2 dynamics are then decoded into target expression. In this manuscript, the authors focused on the decoding step, i.e., from Msn2 dynamics to promoter output. In all the analysis, the authors have assumed that Msn2 dynamics always follows the input chemical signal in every cell and thus the input TF signal equates the external chemical waveforms. However, I feel this assumption could present a fundamental problem in the analysis since it is hard to imagine that every cell in the population has the same Msn2 dynamics under a defined chemical waveform. More specifically, the distribution of responses (YFP level) could likely arise from the heterogeneous Msn2 dynamics among a population. Therefore, I feel that uncoupling the heterogeneity of Msn2 dynamics seems necessary for understanding how Msn2 dynamics contribute to the extent of information transduction*.

We thank Reviewer 2 for bringing this important point to our attention. Since Reviewer 1 raised the same concern, we have addressed this concern in the response to Reviewer 1 above. Furthermore, we have now quantified information loss from 1-NM-PP1→Msn2 for both AM and FM and show this in the new Figure 3—figure supplement 1. We estimate that *I*_AM_(Msn2; 1-NM-PP1)≥2.06 bits and *I*_FM_(Msn2; 1-NM-PP1)≥2.23 bits, but note that these are challenging measurements subject to measurement noise. The values given are therefore likely underestimates. Most importantly, however, our algorithm allows us to correct for information loss from 1-NM-PP1→Msn2 and determine what *I* would have been, had there been no information loss from 1-NM-PP1→Msn2.

*2) Following the point above, the observed difference in signal transmission capacity between AM and FM Msn2 input may likely due to the difference in the degree of heterogeneity of Msn2 dynamics. For example, Msn2 may not follow the FM chemical signal as faithfully as the AM chemical signal. Thus, Msn2 dynamics could be more heterogeneous among cells in FM condition than in AM condition, leading to a more faithful signal transmission for AM. One might need to compare the variability in Msn2 response between AM and FM in order to study the role of Msn2 dynamics in the extent of information transmission in these conditions*.

This is an important point and we thank Reviewer 2 for bringing this issue to our attention. To assess the loss, we have now measured information loss from 1-NM-PP1→Msn2 (*I*(Msn2; 1-NM-PP1)) for both AM and FM. We get *I*_AM_(Msn2; 1-NM-PP1)≥2.06 bits and *I*_FM_(Msn2; 1-NM-PP1)≥2.23 bits. Thus, information loss from 1-NM-PP1→Msn2 is higher for AM than for FM, but *I*_FM_(YFP; 1-NM-PP1) is smaller than *I*_AM_(YFP; 1-NM-PP1) when it comes to gene expression for the four promoters. Together, this shows that the higher promoter information capacity for AM than for FM is not an artifact of higher information loss from 1-NM-PP1→Msn2 under FM than AM, but a real result.

Finally, we would also like to emphasize that even if there had been higher information loss from 1-NM-PP1→Msn2 under FM than AM, this would affect the CFP and YFP gene expression reporters in a correlated manner. Therefore, we correct for this when we apply our algorithm and calculate *I*_int_(YFP; 1-NM-PP1). Since *I*_AM,int_ also always exceeds *I*_FM,int_, we believe that our conclusion, that *I*(YFP; 1-NM-PP1) is higher for AM than for FM for the four promoters studied here, holds.

However, we thank Reviewer 2 for pointing this out and we now show example traces of Msn2 dynamics under FM signals in a new figure (Figure 3—figure supplement 1). We also provide full details on the *I*(Msn2; 1-NM-PP1) calculations and experiments for both AM and FM in the new Figure 3—figure supplement 1.

*3) It occurred to me that the extent of information transduction positively correlates with the dynamic range of promoter response (*Figure 2*). In other words, by simply looking at the top rows (i.e., dose response curves) of each figure panel in*
Figure 2*, I can immediately tell which condition transmits the most information (i.e., mut B AM). Is there any underlying principle that results in such a correlation? This correlation suggests that the dynamic range of the measurement may somehow affect the calculation of mutual information. A potential way to test if this is the case is to characterize the mutual information of the same condition under different lamp power or camera gain settings*.

We agree with Reviewer 2 that there is a strong dependence on dynamic range as also mentioned in the original submission and Figure 2—figure supplement 1. Considering a dose-response, there are three factors that determine *I*: the dynamic range, the noise level and the dose-response shape. The dynamic range and noise level are related – what really matters is the signal-to-noise ratio. It is well-established that noise decreases with increasing expression level (4; 50) and we also observe this (Figure 2—figure supplement 1). Therefore, since the signal-to-noise ratio increases with dynamic range, we generally see a positive relationship between *I* and dynamic range.

The other important factor is the shape of the dose-response. For example, after saturation no further inputs can be distinguished. Therefore, the more linear the dose-response, the higher *I* will be.

As the reviewer suggests, measurement noise is more of a concern at low expression. However, as we show in Figure 2—figure supplement 2, we can accurately measure YFP expression even at quite low expression levels. Furthermore, in our previous paper we showed that *DCS2* exhibits 2-fold lower noise than *SIP18* (supplemental figure S6B in that paper) even when *SIP18* expression is almost 2-fold higher than *DCS2* (Figure 3) (30). Since these experiments were performed on the same microscope with the same LED excitation settings, this shows that the higher noise observed for lower expressed genes with lower dynamic range in this study is not an artifact of our measurement system (such as lamp power or LED intensity).

*4) Regarding the calculation of the mutual information, maximum YFP level was used. The authors made the argument that final protein level is a biologically relevant quantity, which I agreed. In section 2.5 of*
[Supplementary-material SD2-data]*, the authors made arguments about why other quantities are less desirable. I think the authors may want to make the argument more quantitatively. It could be that maximum YFP is the least noisy quantity and thus most suited for calculating mutual information. Such argument can be supported by comparing the CV of possible quantities, such as rate of YFP production, max YFP level, YFP level at a chosen time, etc.*.

We thank Reviewer 2 for their suggestion and we have performed this analysis. We have extended the discussion in section 2.5 and 2.6 of [Supplementary-material SD2-data], where we now also add a Figure showing what *I*_AM_ would have been, had an alternative YFP quantification method been used. Briefly, we find that using the YFP value at a specific timepoint instead of the max YFP value has almost no effect on the *I*_AM_ calculation. Conversely, we find that using the YFP production rate for the calculations leads to a serious underestimation of *I*_AM_. We have now also mentioned this issue in the Materials and methods section.

*5) I am not sure if the authors performed control experiments (in this or previous papers) to show that all the YFP expression (from promoters studied) comes from Msn2 alone (i.e., no other regulators involved). In other words, deletion of Msn2 abolishes the promoter expression under 1-NM-PP1*.

This is a crucial point and we thank Reviewer 2 for pointing this out. We have previously performed precisely the control experiment that Reviewer 2 requests (30). We have the following evidence for specificity in gene expression:

A) In response to 3 μM 1-NM-PP1, both *SIP18* and *HXK1* are strongly transcriptionally upregulated (> 10-fold) in a strain with Msn2 as measured by microarrays. But in an *msn2Δ* strain without Msn2, neither *SIP18* nor *HXK1* changes in expression (Supplementary Figure S1C in (30)). Thus, Msn2 is required for induction of *SIP18* and *HXK1* under 1-NM-PP1 exposure.

B) According to genome-wide Msn2 ChIP experiments, Msn2 binds the promoters of both *SIP18* and *HXK1* (37).

C) In the case of *SIP18*, if we mutate the two Msn2 binding sites (STREs 5′-CCCCT-3′) at positions -386 bp and -367 bp in the promoter, YFP expression decreases at least 20-fold. This shows that the Msn2 binding sites are required for gene activation during 1-NM-PP1 exposure.

We now write that both *HXK1* and *SIP18* are specific target genes of Msn2 in the main text.

[Editors' note: the author responses to the re-review follow.]

*Specifically, the appeal letter now includes the new experiment suggested by the reviewers, in which the information loss between the input and MSN2 dynamics is measured. As the reviewers originally worried, this information loss is quite substantial. This is a bit worrisome, especially as this value is inconsistent with lower loss inferred from the instrinsic/extrinsic analysis (*Figure 3*). Another issue is that this loss was measured only for the AM signal, and not from the FM signal*.

As requested, we have performed additional experiments to quantify information loss for the FM signal. We estimate that *I*_FM_(1-NM-PP1; Msn2) = 2.23 ± 0.03 bits for FM. We provide full details in the new Figure 3—figure supplement 1 which is also shown below; the experiments and analyses were carried out in a manner analagous to the AM calculation described in our previous letter to you. Therefore, information loss due to variability in Msn2-dynamics is not higher for FM than for AM. Most importantly, however, although the input information loss is significant for both AM and FM, we can calculate what *I* would be in the absence of 1-NM-PP1→Msn2 information loss using our algorithm in Figure 3 (*I*_int_).

Regarding the inconsistency referred to by the reviewers, we emphasize that the *I*_FM/AM_(1-NM-PP1; Msn2) analysis quantifies information loss from 1-NM-PP1→Msn2. Our intrinsic/extrinsic algorithm estimates information transfer from Msn2→YFP in the idealized scenario where there is no information loss from extrinsic variables (which include variability from 1-NM-PP1→Msn2, cell cycle phase, ribosome abundances etc.). Therefore, we do not believe that there should be a direct correspondence between the *I*_FM/AM_(1-NM-PP1; Msn2) analysis and the Figure 3 analysis.

This may call for a major rethinking of how to interpret the results in the paper, and in this context, the reviewers suggested the following:

*1) Change some of the interpretation of their data and present the paper with a careful analysis of the information loss due to intrinsic and extrinsic noise sources. This way, the fact that there is substantial loss between chemical input and MSN-dynamics is not a problem anymore, rather an interesting result. Between that and the comparison of AM/FM and the different mutants with increased dynamic range, there should be enough there for an interesting paper. The inconsistencies of the two methods would need to be addressed of course*.

Recent studies have used alternative definitions of “intrinsic” and “extrinsic” noise in characterizing multivariate signaling dynamics (58). We have used the “gene-centric” definitions introduced by Elowitz (24) and we have re-written large sections of the manuscript to emphasize the analysis of information loss due to “gene-intrinsic” and “gene-extrinsic” sources. In the Elowitz definition, “extrinsic noise” is any noise that affects the equivalent CFP and YFP reporters in a correlated manner, such as variability in Msn2 abundance or ribosome abundance. In other words, extrinsic noise comes from cell-to-cell variability in the shared intracellular environment. With our algorithm we can correct for extrinsic noise (including 1-NM-PP1→Msn2 information loss), and with the analysis of *I* for multiple promoter copies (Figure 4) we can estimate how reducing intrinsic noise increases *I*. It is not necessarily the case that *I* without gene-intrinsic noise would equal *I* without gene-extrinsic noise. We have followed suggestion 1 and rewritten the manuscript accordingly.

Furthermore, based on the previous comments, we have also made the following changes to the manuscript:

A) We no longer use the “number of distinguishable states” interpretation and we have removed all instances of “binary”, “ternary”, etc. We have completely revised how we discuss the interpretion of “bits”.

B) We have removed Figure 5, Figure 4 and introduced a new Figure 4 as requested by the reviewers.

C) We no longer say “dynamics” but restrict our claims to AM and FM input as suggested by Reviewer 1. We have also changed the Title, Abstract etc. accordingly.

D) We have revised the discussion of “signal identity vs. intensity” transduction as requested by the reviewers.

E) We have performed the suggested additional analysis including: 1) calculating the joint *I*(CFP, YFP; input) as suggested by Reviewer 1; and 2) calculating *I* using different YFP measures (max value, production rate, value at specific timepoint etc.) as suggested by Reviewer 2.

F) We have included evidence that the transcriptional response of *SIP18* and *HXK1* are specific to Msn2 as suggested by Reviewer 2.

*2) Quantify more directly the information transmission capacity between MSN2 dynamics and MSN2 promoter by measuring in the same cell MSN2 the dynamics of localization in the nucleus and the resulting promoter reporter. Since mutual information is a symmetric quantity, one could bin the promoter response into 8 or 16 bins. Than calculate the mutual information between the scalar MSN2 promoter response and the distribution of multivariate dynamics responses in each bin. This could be done by pooling all the chemical input data together and use an approach such as described e.g. in*
[58]
*to calculate mutual information between scalar input and dynamic response. Perhaps the data in*
Figure 2—figure supplement 2
*is sufficient, or if not, additional experiments are required*.

We performed the suggested analysis. We find that Msn2-mCherry measurement noise during the full time-lapse experiments is too high for us to gain insight from this analysis. We therefore cannot include this analysis in the manuscript. For reference, we describe below how we performed the analysis.

For all experiments, we measured Msn2-mCherry dynamics and YFP+CFP reporter expression in single cells at 2.5 min time-resolution without collecting a finely-spaced z-stack series. As mentioned in our previous letter, suggestion 2 is technically challenging for two reasons. First, the nucleus moves in-and-out of focus during time-lapse acquisition which makes accurate quantification challenging. Second, to partially overcome this problem, z-stack images through the nucleus must be acquired, which causes high photobleaching and makes time-lapse imaging challenging. For example, to accurately quantify Msn2-dynamics during FM, it is necessary to image with 1-min time resolution and perform a finely-spaced z-stack series with long exposure times (as performed in new Figure 3—figure supplement 1). However, such intense imaging causes too severe photobleaching within 20-25 frames to enable accurate quantification Therefore, measurement noise in nuclear Msn2-mCherry quantification makes the suggested approach challenging.

Nevertheless, we have performed this analysis using two approaches:

Approach 1: We pooled all data for a given promoter (e.g. *HXK1* for AM input). We bin data based on the measured Msn2-mCherry amplitude in each cell. Then we calculate *I*_AM_(YFP; Msn2-mCherry).

Approach 2: For each condition (e.g. 100 nM 1-NM-PP1 for AM input), we removed cells where the measured Msn2-mCherry amplitude was outside the mean±10% range. That is, for each 1-NM-PP1 input, we removed outlier cells with too high or too low measured Msn2-mCherry amplitude relative to the mean. Then we calculate *I*_AM_(YFP; 1-NM-PP1) using the trimmed data set.

With both approaches, we find an *I*_AM_ that is nearly identical, within error, to the *I*_AM_ we calculate in Figure 2 (e.g *I*_AM_ for *HXK1* is 1.30 (Figure 2), but 1.29 bits using approach 2). Since variability in Msn2-mCherry almost certainly contributes to YFP expression variability, this indicates that the time-lapse Msn2-mCherry measurements suffer from too high measurement noise to meaningfully interpret these observations.

Finally and most importantly, we would like to emphasize that our algorithm allows us to estimate the maximal information transduction capacity of a promoter—and this is the focus of our manuscript—in an idealized cell without variability in Msn2 dynamics and abundance. In conclusion, suggestion 2 asks us to condition the YFP response on measured Msn2 variability. Our algorithm achieves this goal by estimating the YFP response in an idealized cell without Msn2 variability.